# Sensitivity of transatlantic dust transport to chemical aging and related atmospheric processes

Mohamed Abdelkader[1,5,*], Swen Metzger[1,2,3], Benedikt Steil[1], Klaus Klingmüller[1], Holger Tost[4], Andrea Pozzer[1], Georgiy Stenchikov[5], Leonard Barrie[6], and Jos Lelieveld[1,2]

[1]Max Planck Institute for Chemistry, Mainz, Germany
[2]The Cyprus Institute, Nicosia, Cyprus
[3]Eco-Serve, Freiburg, Germany
[4]Johannes Gutenberg University, Mainz, Germany
[5]King Abdullah University of Science and Technology, Saudi Arabia
[6]Stockholm University, Stockholm, Sweden
[*]This work has been performed at the The Cyprus Institute.

*Correspondence to:* M. Abdelkader (m.abdelkader@mpic.de), S. Metzger (s.metzger@cyi.ac.cy)

**Abstract.** We present a sensitivity study on transatlantic dust transport, a process which has many implications for the atmosphere, ocean and climate. We investigate the impact of key processes that control the dust outflow, i.e., the emission flux, convection schemes and the chemical aging of mineral dust by using the EMAC model following Abdelkader et al. (2015). To characterize the dust outflow over the Atlantic Ocean, we distinguish two geographic zones: (i) dust interactions within the ITCZ (DIZ) and (ii) the adjacent dust transport over the Atlantic Ocean (DTA). In the latter zone, the dust loading shows a steep and linear gradient westward over the Atlantic Ocean, since particle sedimentation is the dominant removal process, whereas in the DIZ zone aerosol–cloud interactions and wet deposition / scavenging processes determines the extent of the dust outflow. Generally, the EMAC simulated dust compares well with CALIPSO observations, however, our reference model configuration tends to overestimate the dust extinction at lower elevation and underestimates it at higher elevation. The Aerosol Optical Depth (AOD) over the Caribbean responds to the dust emission flux, only when the emitted dust mass is significantly increased over the source region in Africa by a factor of ten. These findings point to the dominate role of dust removal (especially wet deposition) in transatlantic dust transport. Experiments with different convection schemes indeed revealed that the transatlantic dust transport is more sensitive to the convection scheme than to the dust emission flux parameterization.

To study the impact of dust chemical aging, we focus on a major dust-outflow in July 2009. We use the calcium cation as a proxy for the overall chemical reactive dust fraction and consider the uptake of major inorganic acids (i.e., $H_2SO_4$, $HNO_3$, HCl) and their anions, i.e., sulfate ($SO_4^{2-}$), bi-sulfate ($HSO_4^-$), nitrate ($NO_3^-$) and chloride ($Cl^-$)) on the surface of mineral particles. The subsequent neutralization reactions with the calcium cation forms various salt compounds that cause the uptake of water vapour from the atmosphere, i.e., by chemical aging of dust particles leading to an increase of 0.15 in AOD under subsaturated conditions (monthly mean, July 2009). As a result of the radiative feedback on surface winds, dust emissions regionally increased. On the other hand, the aged dust particles, compared to the "non-aged" case, are more efficiently removed by both wet and dry deposition, due to the increased hygroscopicity and particle size (mainly due to water uptake). The enhanced removal of aged particles decreases the dust burden and lifetime, which indirectly reduces the dust AOD by 0.05

(monthly mean). Both processes can be significant (major dust-outflow, July 2009), but the net effect depends on the region and level of dust chemical aging.

# 1   Introduction

In the past several decades, transatlantic dust transport has gained tremendous attention because of many important impacts on Earth's climate, human health and ecosystems. North African dust transport over the Atlantic Ocean has emerged as a major contributor to the soil nutrient input to many islands in the Caribbean, the Bahamas (Muhs et al., 2007), Bermuda (Muhs et al., 2012) and in the Amazon Basin (Bristow et al., 2010; Ben-Ami et al., 2012; Abouchami et al., 2013). Dust deposition influences the oceanic and terrestrial biogeochemistry by the transport of nutrients such as iron (Ussher et al., 2013; Baker et al., 2013, 2010; Jickells et al., 2005) and phosphorus (Nenes et al., 2011) that dissolve into the ocean water. The emission, transport, and deposition processes of the North African dust are strongly influenced by meteorology causing strong seasonal, inter-annual and decadal variability (Mahowald, 2007; Mahowald et al., 2010). Large fractions of the dust emissions are carried across the west coast of North Africa up to the Western Atlantic (Prospero et al., 2014) and significant correlations exist between the dust and climate variables, such as sea surface temperature, the North Atlantic Oscillation (NAO), and the Madden-Julian Oscillation (MJO) (Ginoux et al., 2004; Wong et al., 2008; Guo et al., 2013). In addition, the African dust in the Sahara air-layer region influences the rates of rainfall in the Inter-Topical Convergence Zone (ITCZ) (Huang et al., 2009, 2010), and its radiative impacts can shift and widen the ITCZ northward (Bangalath and Stenchikov, 2015).

African dust is transported in great quantities to the Caribbean basin throughout the year, although the strong seasonal cycle shows the maximum transport of dust in boreal summer and the minimum in winter (Prospero et al., 2014; Yu et al., 2015). The seasonality is corroborated by satellite measurements of Aerosol Optical Depth (AOD), which show extensive plumes of high AOD in summer extending from the west coast of Africa to the Caribbean, the Gulf of Mexico, and to the southern United States (Hsu et al., 2012; Yu et al., 2013; Chin et al., 2014; Kim et al., 2014; Groß et al., 2015). The satellite data also indicate that the dust transport to the Western Atlantic in winter and spring is comparable, but the dust is largely confined to the southern latitudes of Barbados with a plume axis crossing the coast of South America in the region of French Guiana and Surinam. In addition, satellite data indicate a decrease of 50% in AOD and a decrease of 0.1–0.2 in the dust-only optical depth during the transport (Kim et al., 2014). The ITCZ acts as an efficient removal mechanism (Prospero et al., 2014) and thus as a barrier to the transport of dust to the southern Atlantic (Huang et al., 2009, 2010; Adams et al., 2012). To characterize the transatlantic dust transport, many studies have used satellite observations (Liu et al., 2008; Ben-Ami et al., 2009, 2010; Adams et al., 2012; Ben-Ami et al., 2012; Ridley et al., 2013; Alizadeh-Choobari et al., 2014; Kim et al., 2014; Yu et al., 2015, among others). However, the estimation of the satellite-based dust flux has large uncertainties, primarily because of ambiguity associated with the derived dust-only optical depth (Yu et al., 2009, 2013) and the dust mass extinction efficiency. Both parameters are used for calculating the dust mass loading (Kaufman, 2005).

One cause of uncertainty is the chemical aging of mineral dust. For instance, the condensation of inorganic acids, such as nitric acid ($HNO_3$), can alter the particle size due to changes in hygroscopicity of the dust particles (Metzger et al., 2006;

Karydis et al., 2016). $HNO_3$, which is an oxidation end product of combustion processes and lighting $NO_x$, and therefore ubiquitous in the atmosphere, readily reacts with the calcium of the mineral dust surface. The neutralization product, calcium nitrate, additionally takes up ambient water vapour, which can change the particle (wet) radius. This process of water uptake can become significant, since it already starts at a relative humidity as low a 50% (the relative humidity of deliquescence (RHD) of $Ca(NO_3)_2$ is 48% at T=298 K). In strong contrast, dust coating by sulphuric acid ($H_2SO_4$) does not lead to such hygroscopic particles, since the RHD of $CaSO_4$ is close to 100% (at any T). Thus, especially the coating by nitrates can determine the hygroscopicity of mineral dust particles in case of a polluted atmosphere (Bauer et al., 2007; Sullivan et al., 2007; Li and Shao, 2009; Tobo et al., 2009, 2010; Li et al., 2013). The growth of the particles increases the scattering cross sections and therefore alters the AOD, it indirectly affects the cloud scavenging efficiency (Lance et al., 2013; Wu et al., 2013; Li et al., 2013), overall potentially increasing the wet and dry removal of the dust particles (Abdelkader et al., 2015).

Therefore, the dust cycle and the associated impacts are found to be challenging for global and regional modeling, because the complex dust processes have to be parameterized using a suite of simplifications (Astitha et al., 2010; Nowottnick et al., 2010; Huneeus et al., 2011; Ridley et al., 2013; Kim et al., 2014; Gläser et al., 2015). Although most sophisticated atmospheric models can reproduce the transatlantic dust transport plumes, the patterns differ in magnitude and seasonality. Generally, the models show better performance in summer than in winter for the transatlantic dust transport (Huneeus et al., 2011). It has been observed that large uncertainties particularly exist between model simulations of the dust deposition (wet and dry) (Schulz et al., 2012). The atmospheric models that are applied in the AeroCom model intercomparison activity (http://aerocom.met.no/) show that the mean normalized bias of the AOD varies within a wide range from –0.44 to 0.27 (Huneeus et al., 2011), which is caused by large discrepancies in the dust-related processes (emission, horizontal and vertical distributions and the parameterization of chemical aging) that affect the dust transport from Northern Africa over the Atlantic ocean (Prospero et al., 2010). This indicates that in these models the dust removal is very efficient during the transatlantic transport (Kim et al., 2014) and that the development of the model requires comprehensive representation of the dust related processes. Though the incorporation of satellite products helps improving the model results, a deeper understanding of the key factors that determine the transport of the dust is also required. This study aims at examining the factors that can affect the transatlantic dust transport, i.e., the emission flux, convection schemes and the chemical aging of mineral dust, by using the EMAC model.

## 2   Model Description

We use the EMAC (ECHAM5/MESSy2 Earth System Model) following Abdelkader et al. (2015). The EMAC model describes the tropospheric and middle atmosphere processes and their interactions with land and oceans considering various submodels (Joeckel et al., 2010) – those used in this study are listed in Table 1. The mineral dust particles are emitted in two log-normal distribution modes (accumulation and coarse) with median diameters of 0.5 μm and 5.0 μm and a modal standard deviation of 1.59 and 2.0 for the accumulation and coarse modes respectively (Abdelkader et al., 2015). The anthropogenic emissions are based on the EDGARv4.0 inventory (Pozzer et al., 2012) and includes the greenhouse gases, $NO_x$, CO, non-methane volatile organic compounds (NMVOCs), $NH_3$, $SO_2$, black carbon (BC) and organic carbon (OC) from fossil fuel and biofuel

use. The monthly large-scale biomass burning emissions of OC, BC and $SO_2$, are based on GFED version 3 (Global Fire Emissions Database) (van der Werf et al., 2010). The emissions drive a comprehensive atmospheric chemistry mechanism (Sander et al., 2005), which calculates major inorganic acids ($H_2SO_4$, $HNO_3$, $HCl$) online with meteorology. Organic acids are not considered in this model setup since their concentrations over Sahara during dust outflow are very low, though, many

modeling studies reported the uptake of organic acids by dust particles (Metzger et al., 2006; Möhler et al., 2008; Liu et al., 2013; Li et al., 2013; Alexander et al., 2015; Wang et al., 2015).

The chemical aging of the dust depends on the condensation of inorganic acids and the associated uptake of water vapor. This increases the dust particle mass, particle size and the removal rates, which tends to decrease the lifetime of chemically aged dust. The condensation of acids in our model yields the anions sulfate ($SO_4^{2-}$), bi-sulfate ($HSO_4^-$), nitrate ($NO_3^-$), and chloride

($Cl^-$), whereas the condensation of ammonia ($NH_3$) yields a semi-volatile cation, ammonium ($NH_4^+$), that reacts with the inorganic anions in competition with the mineral cations $Na^+$, $Ca^{2+}$, $K^+$, $Mg^{2+}$ (Metzger et al., 2006). However, in this study the cations are considered as reactivity proxy for natural aerosols, such as sea salt, biomass burning, or mineral dust, where we follow Abdelkader et al. (2015) and use a fixed percentage. These fractions have been derived from a comprehensive sensitivity study (which will be presented in a separate study) to achieve the best agreement of the cation and anion concentrations

with various station observations for the period 2000-2012 (see Section 3). The anion–cation neutralization products (salt compounds), simulated by the aerosol thermodynamic models, ISORROPIA-II (Fountoukis and Nenes, 2007) or EQSAM4clim (Metzger et al., 2016), can alter the hygroscopicity of the atmospheric dust particles, but the effect strongly depends on the atmospheric residence time, region and concentrations of acids. Generally, dust chemical aging changes the solubility, which controls the water uptake and in turn alters the aerosol size distribution (Metzger et al., 2006). The water uptake is a key

parameter and important for aerosol-radiation feedback, aerosol in-cloud processing (nucleation scavenging), and below-cloud (impaction) scavenging. The EMAC scavenging processes include detailed pH-dependent aqueous phase chemistry (Tost et al., 2006a) which is fully coupled with the aerosol and gas-phase chemistry, liquid cloud water and ice crystals. In addition to the aerosol hygroscopic growth and scavenging, the dust size distribution can change by coagulation, and smaller particles can grow into larger sizes for both the soluble and insoluble aerosol modes (Pringle et al., 2010), whereas aerosol hygroscopic

growth is only allowed in the soluble modes (Abdelkader et al., 2015). Dry deposition and particle sedimentation can remove all particles from the atmosphere depending on the particle size (Kerkweg et al., 2006a). Thus, the representation of the dust cycle in our EMAC setup couples the dust emissions, loading, and lifetime with the radiative forcing and model dynamics. As a result, changes in the dust loading feed back to the surface wind speed, soil moisture, cloud formation and precipitation, and in turn the dust emission flux. Overall, the level of air pollution controls the dust cycle because it determines the level of dust

chemical aging by inorganic acids and water vapor. A Newtonian relaxation approach is used to nudge the model meteorology in the free atmosphere (i.e., above the boundary layer) to achieve a realistic simulation of the surface wind speed and tracer transport (Abdelkader et al., 2015). Nudging significantly improves the surface dust mass concentration over the Caribbean compared with dust observations (Astitha et al., 2012). The model spectral resolution is T106 ($\approx 110$ km) and for the longterm simulations it is T42 ($\approx 280$ km). Both model resolutions use 31 vertical levels. Figure 1 summarizes the representation of the

dust cycle and air-pollution-dust-chemical-aging-radiation feedbacks in our EMAC model setup.

# 3 Long-term evaluation

This study aims at examining the key factors that affect the transatlantic dust transport for a major dust-outflow event in July 2009 with a model resolution of T106, which is presented in Section 4. Before we focus on the sensitivity study, we present in this section the key findings of a comprehensive model evaluation, which was performed for the period 2000–2012 with a coarser resolution of T42. For the long-term evaluation, we use the following satellite and ground station AOD products:

- AErosol RObotic NETwork (AERONET) – Holben et al. (1998);

- Cloud-Aerosol Lidar and Infrared Pathfinder Satellite Observations (CALIPSO) – Winker et al. (2009, 2007);

- MODerate resolution Imaging Spectroradiometer (MODIS) platforms Aqua and Terra
  (product collection 6, L3 gridded data) – Kaufman et al. (1997);

- Precipitation data from Tropical Rainfall Measuring Mission (TRMM)
  (product version 31 L3 gridded data) – Diner et al. (1998);

Figure 2 shows the seasonal average of the simulated dust burden and the precipitation rate over a 13-year simulation period. Both the dust burden and the precipitation rate peak during the summer season (JJA), where the dust plume is located relatively far north from the equator, in agreement with remote sensing observations (Prospero et al., 2014; Yu et al., 2015). During the winter season (DJF), the dust burden and the precipitation rate show a minimum, whereas during the spring season (MAM), the dust plume and the ITCZ are shifted southward. In winter and spring, the dust transport shifts southward to 0°-10°N and affects South America significantly, whereas during summer, the dust transport occurs predominantly at 10°-20°N, substantially affecting the Caribbean (Yu et al., 2015). During boreal winter the enhanced precipitation over the Northern part of South America results in higher and localized dust scavenging because the precipitation along the dust transport from the Western Africa into the Caribbean is at minimum. In contrast, during boreal summer, the dust spreads to a larger extent into the ITCZ because of the stronger emissions (Prospero et al., 2014) while it is subject to enhanced dust scavenging. The strong southward gradient of the dust burden ($\approx 100 \, \mathrm{mg \, m^{-2} \, deg^{-1}}$) is collocated with precipitation in the western part of the Sahel and the ITCZ region. During the winter months, dust is primarily scavenged over Southeast America. As a result, the extent of the dust outflow is primarily controlled by precipitation in the ITCZ region. Figure S1 in the Supplement shows the dry and the wet removal of the dust particles. It shows that the dry removal dominates the northern part of the dust outflow region, whereas the wet removal dominates the southern part.

To indicate the region where the dust interacts with the ITCZ, we introduce the dust-ITCZ (DIZ) zone which is shown in Figure 3 – the DIZ is marked by a blue line, and the AERONET station locations used to evaluate the simulated AOD are included. In the DIZ region, the transatlantic dust transport is controlled by dust–cloud interactions and the dust scavenging is most efficient. Accordingly, we refer to the region of the pre-dominant dry removal process (sedimentation), as the DTA zone.

Summarizing the long-term evaluation results, Figure 4 shows: (i) the transatlantic dust transport region with skill score (Taylor, 2001) at each station (see Appendix A for the evaluation metrics), (ii) the time series of the six selected stations

that provide long-term data with three stations each in the Caribbean (left) and around West Africa (right), and (iii) the corresponding scatter plots of both sides of the Atlantic Ocean include the observations from all stations. Table 2 summarizes the model performance for both regions over the entire period (2000–2012) for all stations. The 13-year average (based on 5 hourly output) of the simulated AOD for the Western Africa sites is 0.16±0.27 (one standard deviation), which is lower than the observation of 0.24±0.37. The difference is larger compared to that for the Caribbean, for which the average simulated AOD is 0.12±0.18 and 0.14±0.22 according to the observations. At both sides of the Atlantic, the lower variability of the model is primarily a result of the relatively coarse model resolution (T42≈280 km) (Gleser et al., 2012), which was used for these long-term simulations because of significantly larger computational burden for the higher T106 resolution. The skill score (SS1) has a value of 0.73 and 0.70 for the Western Africa and the Caribbean stations, respectively.

Additionally, the correlation coefficients (R) (Table 2) are lower than the SS1, because R is more phase sensitive than the SS1 (i.e., more sensitive to time lags between simulated and observed AOD). The higher R value for West Africa (0.61) compared with Caribbean (0.41) mainly results from the overall higher contribution of dust AOD to the total AOD. Typically, the Caribbean is strongly influenced by the uncertainty associated with long-range transport and the dust chemical aging, with potential failures causing a time shift of dust peaks during the transport. These differences are, however, best revealed by station time series (Figure 4). The six stations are selected based on the availability of observations during the period 2000-2013, while the other stations have a significantly lower number of observations and therefore lower skill scores. The comparison shows that the model captures the variability of AOD at all stations, and only around West Africa the model underestimates the AOD peaks, especially at Dakar which is at the edge of the DIZ zone. Over the Caribbean, the model generally underestimates the AOD during the dust outflow periods, e.g., seen at the AERONET station La Parguera. This underestimation could be related to the representation of dust emissions and related processes in the source region of the West Africa (Huneeus et al., 2011; Shao et al., 2011; Cuevas et al., 2015), by overestimated removal during transport (Schulz et al., 2012; Prospero et al., 2014), or due to low-biased dust transport from the boundary layer into the free atmosphere (Khan et al., 2015). In addition, the underestimation of AOD could be also due to the missing fraction of giant mode particles (larger than $10\,\mu m$), which may contribute to an underestimation of AOD near the dust source region. However, giant particles are not transported far over long distances and hence not really relevant for the long-range transport and our sensitivity study on the emission flux and removal mechanisms.

## 4 Sensitivity studies

To study the key factors that may affect the transatlantic dust transport, we focus in this section on a major dust-outflow event that occurred in July 2009. We study the impact of various key factors with a relatively high model resolution (T106). The dust outflow is, in terms of AOD ,close to most observations, as indicated by monthly means highlighted by the red bar in Fig. 4. However, near the source region at Dakar and Capo Verde, the AOD observations are underestimated for this month. During this period a major outflow event occurred, and therefore it seems suitable to test various model parameters, i.e., (a) the dust emission flux, (b) the convection parameterization, and (c) the level of dust chemical aging.

Figure 5 shows the dust burden and the total mean precipitation for July 2009 from the reference EMAC simulation, which includes the dust cycle and chemical aging as shown in Fig. 1. The simulated dust surface concentration reaches on average up to $600\,\mu\mathrm{g\,m}^{-3}$ at Dakar, indicating that the model captures in principle the strong outflow event. Generally, two strong precipitation areas are visible with one peak centered at 15°W with a monthly average of $20\,\mathrm{mm\,day}^{-1}$, i.e., one at the coast of West Africa and the other peak area is located in the Caribbean at 50°W. with a monthly average of $25\,\mathrm{mm\,day}^{-1}$. These precipitation maxima influence the dust loading. During transatlantic dust transport, the ITCZ represents a strong barrier for the dust outflow and therefore controls the meridional extent of the dust plume (Yu et al., 2015). The ITCZ acts as a major sink that depends on the amount of precipitation (Prospero et al., 2014; Schlosser et al., 2014) and the removal might be enhanced depending on the dust chemical aging (Abdelkader et al., 2015). Clearly, the precipitation within the ITCZ coincides with the steep gradient of the dust burden in the meridional direction over the Western Africa. Along the zonal extent of the dust plume, the collocation of the dust plume and precipitation corroborates that the meridional extent of the dust is primarily controlled by the location of the ITCZ. Fig. S1 in the Supplement summarizes the monthly average dust removal during July 2009. Table 1a and table 1b in the supplement additionally show some evaluation metrics for the AOD of the sensitivity study over the West African and Caribbean stations.

Typically, African dust outflow reaches the Caribbean $\approx 5$ days later (Gläser et al., 2015) and the surface dust concentration is significantly lower at the Caribbean side compared to Western Africa. Figure 6 shows the time series of the size-resolved surface dust concentrations. Two main dust outflows on the $2^{nd}$ and $12^{th}$ July are simulated at the Capo Verde station, indicated by dust concentrations higher than or close to $300\,\mu\mathrm{g\,m}^{-3}$ (equivalent particle cutoff diameter of $5\,\mu\mathrm{m}$) and another weaker dust outflow is simulated on $24^{th}$ July, indicated by a lower concentration peak around $100\,\mu\mathrm{g\,m}^{-3}$. The former two dust outflows are seen at Dakar with twice the concentration (up to $600\,\mu\mathrm{g\,m}^{-3}$) at slightly different time periods due to different transport. Eventually, the dust outflow reaches the Caribbean with a significant lower concentration of around $60\,\mu\mathrm{g\,m}^{-3}$ at the earth surface.

Despite chemical aging, the model simulates a majority of the dust particles in the insoluble coarse (ci) mode, which indicates that the dust particle concentration is high and/or the inorganic acids concentration is relatively seen too low for complete chemical aging. This is especially valid for strong dust outflows, such as studied here. On the other hand, the fraction of the aged dust, i.e., the ratio of the coarse mode soluble to insoluble particles (cs/ci), is somewhat higher in the Caribbean because of the continuous chemical aging during long-range transport. The aged dust fraction over West Africa is about 10% of the total dust mass and twice that at the Caribbean sites. The same is true for the dust in the accumulation modes (ai and as), but the mass concentrations are an order of magnitude lower compared to the coarse mode concentrations and therefore they are not discernable at the linear scale. At higher elevations, this fraction can be different because of different dust and precursor gas concentrations.

To investigate the vertical distribution, the simulated dust extinction is compared with the dust subtype classification of the CALIPSO retrievals. Figure 7 shows a comparison for the second dust outbreak on 12 July 2009. The figure shows a subset of four collected CALIPSO tracks and includes a qualitative comparison of the dust layer height. The scatter plot attached to each panel represents the point-to-point comparison, colored by the height of each observation point whereas the area plots

show the dust burden interpolated in time to the CALIPSO overpass time, indicated by a solid black line. Additional CALIPSO tracks are shown in Fig. S2a–Fig. S2e in the Supplement. Both EMAC and CALIPSO show that dust over the Sahara reaches an elevation up to 7 km. The dust burden is very low (as indicated by the area plot) south of 10°N, which coincides with a very low AOD observed by CALIPSO. Both EMAC and CALIPSO show that the dust plume is limited to the area between

14°to 22°N and the top of the dust layer is lowered to 5 km over the middle of the Atlantic. This is primarily a result of the prevailing deposition (gravitational settling + wet removal), which is further discussed in the following sections. Once the dust reaches the Caribbean, the plume spreads over a considerably larger area, which extends from 5°to 28°N as a result of changes in meteorological conditions. The dust plume eventually reaches the Caribbean with a top layer height of $\approx 5$ km. In Fig. 7, the comparison with CALIPSO (and Fig. S2a–Fig. S2e in the Supplement) shows that the model captures the vertical

structure of the dust outbreak during the transport over the Atlantic Ocean. Nevertheless, the model tends to systematically overestimate the dust extinction at lower altitudes, whereas at higher altitudes the model tends to underestimate the CALIPSO extinction (considering all CALIPSO tracks in Fig. 7 and Fig. S2a–Fig. S2e in the Supplement). This indicates that EMAC might remove the dust too efficiently during transport. The reason can be manifold and related to different processes of the dust cycle (Figure 1). Therefore, the key factors are investigated further in greater detail.

## 4.1  Dust emission flux

A successful representation of the dust cycle first of all depends on an accurate dust emission flux. However, the simulated emission flux critically depends on many model parameters, where some of them are resolution dependent. Using EMAC, the dust emissions are calculated considering the friction velocity following Astitha et al. (2012). To test the sensitivity of the transatlantic dust transport to the dust emission parameterization, several sensitivity simulations were performed, which are

summarized in Table 3. The total dust mass, emitted during July 2009 within the region between 20°W to 10°E and 15°N to 30°N, is 0.6133 kg m$^{-2}$ for the reference case.

The first test case (B1E1) represents a redistribution of emission bins between the coarse and accumulation modes so that dust particles are shifted from the coarse to the accumulation mode while conserving the total dust mass. In this case, a larger amount of dust in the accumulation mode is transported over extended distances compared with the reference case "EMAC".

"EMAC" considers the same total dust mass with a larger fraction in the coarse mode. Additional sensitivity runs, B1E2 to B1E7, change the total dust emission flux by increasing the emission flux according to different factors shown in Table 3. The horizontal dust emission flux is described by Eq. 1 (Marticorena and Bergametti, 1995; Astitha et al., 2012)

$$H = \frac{c\rho_{air}u_*^3}{g}(1 + \frac{u_t^*}{u^*})(1 - \frac{u_t^{*2}}{u^{*2}}), u^* > u_t^* \tag{1}$$

With the tuning parameter c = 1 representing the reference case "EMAC" following (Darmenova et al., 2009; Astitha et al.,

2012), g is the gravitational acceleration, $\rho_{air}$ the air density, $u^*$ the friction velocity, $u_t^*$ the threshold friction velocity. For case B1E8, the horizontal mass flux is increased by a factor of 2.6 (parameter c in Eq. 1). The cases highlighted in Table 3 are shown in Fig. 8, whereas the other cases are shown in the Supplement (Fig. S3).

Due to the different dry and wet deposition characteristics of the accumulation and coarse mode particles, significant differences are expected. Figure 8 shows that the AOD time series at the selected AERONET stations are rather insensitive to the emission flux modifications except for case B1E3 (and B1E4, which is shown in the Supplement). This is valid for both sides of the Atlantic, where the AOD at the Caribbean stations seems even less sensitive than the AOD for the West African sites. Only for the cases where the coarse mass flux is significantly increased (factor of 5.3), the AOD shows a higher sensitivity. The large increase in the coarse mode mass for case B1E3 results in a significant increase in AOD (exceeding 2.0) on both sides of the Atlantic Ocean. Case B1E8 (modification of the horizontal mass flux) shows better agreement with the AERONET observations at both sides of the Atlantic Ocean despite the very high AOD values obtained on 21 July at Saada station. The model captures the AOD during the two dust outflow events (2 July and 12 July) at Capo Verde as well as the first dust outflow at Saada on 4 July. For the Caribbean sites, case B1E8 shows the best agreement with AERONET for the three stations.

The sensitivity simulations show that the accumulation mode fraction of the dust contributes much less to the AOD on both sides of the Atlantic Ocean because even an increase by a factor of 5.3 in the dust emission flux is not sufficient to match the observations. Instead, such an increase (by a factor of 5.3) in the emitted dust mass flux results–regionally and globally–in an unreasonable dust budget shown by Astitha et al. (2012). On the other hand, this sensitivity study shows that the AOD is more sensitive to the dust mass in the coarse mode and that the AOD over the Caribbean is much less sensitive to the total dust emission flux. Clearly, the model sensitivity is higher for the West African sites because these AOD results are more directly controlled by the Saharan dust outbreaks. To match the elevation at which this outflow occurs is equally important. The comparison with the CALIPSO observations (Fig. 7) reveals that EMAC overestimates the dust extinction at lower elevations, whereas the values at higher elevations are underestimated. This finding points to the strong contribution of dust removal during transatlantic dust transport, and is largely controlled by the convection scheme.

## 4.2 Convection schemes

The scavenging of dust particles by precipitation is another key factor that controls the transatlantic dust transport (Kim et al., 2014). In order to study the impact of the convection and the associated precipitation during the dust outflow, different convection schemes (implemented in EMAC by Tost et al. (2006b)) are compared. The default scheme (TIEDTKE convection with NORDENG closure) provides realistic water vapor distributions on the global scale, which is crucial for radiative transfer processes and atmospheric chemistry (Tost et al., 2006b, 2010; Rybka and Tost, 2013). However, the radiative effect of aerosols has not been considered in these studies. Table 3 includes the sensitivity tests by using several convective schemes available in the EMAC model. The principal cases are shown in Fig. 9, whereas the other cases are shown in the Supplement (Fig. S4).

Figure 9 depicts the AOD time series for the stations shown in Fig. 3 and shows a larger sensitivity to the convection compared to the emission flux parameterizations (Sec. 4.1). In particular, the AOD is more sensitively influenced over the West African than over the Caribbean sites, which is primarily a result of the decreasing dust burden due to the removal of the dust during transport (Fig. 6). Generally, the AOD is underestimated at all stations, except for Saada, in the reference simulation (EMAC). During the period 20–25 July 2009, this significantly improves in the sensitivity simulations (B1T3 and B1T5). However, the model also simulates a dust outflow event that is not observed by the AERONET stations. Overall, over

the Caribbean, case B1T5 (ECMWF operational convection scheme) yields the best results for all dust outflow events. The main differences between the schemes appear in the tropical region, while the maximum difference is obtained during boreal summer. For these conditions (location + time), the EMAC reference setup is associated with relatively large discrepancy in the precipitation amount (Tost et al., 2006b). As a result, the scavenging of aerosols, including dust particles, is overestimated due to the high precipitation rates. Consequently, this over-removal of the dust results in an underestimation of the AOD over the Caribbean.

Figure 10 illustrates this finding. The total cloud fraction, precipitation, dust surface concentration, and the dust burden (monthly mean) are shown for the different convection parameterizations in comparison to MODIS cloud fraction and TRMM precipitation. In general, the model reproduces the main features of the cloud cover observations; however, EMAC (reference) underestimates cloud cover over the Atlantic Ocean. Over the tropical areas in Africa, B1T5 (ECMWF) leads to more realistic results compared to MODIS and compared to B1T4 (also ECMWF but with shallow convection closure, shown in Fig. S5 in the Supplement). Over the ocean, B1T5 considerably underestimates cloud cover and precipitation rates. Over the Caribbean sites, B1T5 overestimates cloud cover, whereas the other schemes produce more realistic results. On the other hand, the calculated precipitation (second column) generally shows an overestimation for all schemes except B1T5 with an underestimation over the ocean. As a result of the differences in the cloud cover and precipitation rates, all model simulations show different magnitudes of the dust plumes (third and fourth columns) which is most pronounced for the dust burden. For the reference simulation (EMAC in Table 3 and Figure 10), the dust plume extends to $60°$W with a dust burden of 200 mg.m$^{-2}$, whereas for simulation B1T3 (TIEDTKE) the same dust burden is obtained at $80°$W and westwards. The difference in the dust plume magnitude merely results from different removal efficiencies because of different precipitation rates.

For a quantitative comparison, the average meridional dust burden in the dust outflow over the Atlantic Ocean region ($10°-25°$N) is shown in Fig. 11 for different convection parameterizations. Additionally, the precipitation and the column averaged aged dust proxy (ADP), which was introduced by Abdelkader et al. (2015), are included. The ADP simulations, which represents the ratio between aged and non-aged dust particles, indicates the level of dust chemical aging (i.e., the mass fraction of the aged to the total dust mass). A zero ADP value indicates "pristine" or freshly emitted insoluble particles (no aging), whereas a value of one indicates that all dust particles are chemically aged (fully coated and transfered from the insoluble to the soluble modes).

First, the dust burden shows a very steep gradient westward over the Atlantic Ocean. This is mainly a result of dust removal by deposition (sedimentation and scavenging mechanisms) during long-range transport. Over the Atlantic (within DTA), this gradient is linear in the logarithmic scale, whereas the gradient is nonlinear over the Western and Eastern Atlantic (especially within DIZ). The dust burden over West Africa (east of $10°$W) is about $1000\,\mu\mathrm{g\,m^{-3}}$ but declines to $50\,\mu\mathrm{g\,m^{-3}}$ over the Caribbean. The different parameterization schemes show more than a factor of 2 difference between the dust burden over Western Africa, and about a factor of 3 over the Caribbean. This is primarily a result of different precipitation rates and the associated differences in dust removal. The two precipitation peaks (over Western Africa and the Caribbean), shown in Fig. 5, are also seen in Fig. 11. They are, however, weaker because the averaging is performed over a wider area (dust plume) that is not associated with precipitation. The higher precipitation rate over the western and eastern parts of the Atlantic results in

enhanced dust scavenging. Over the Atlantic, the precipitation is lower and therefore the removal by sedimentation is stronger during July 2009 ($\approx 2\,\mathrm{g\,m^{-2}}$ compared to $\approx 0.2\,\mathrm{g\,m^{-2}}$, respectively). The elevated precipitation over the Caribbean causes maximum wet deposition. As a result, the dust burden is an order of magnitude lower over the Caribbean compared to West Africa. In addition, there is a clear anti-correlation between the dust burden and the precipitation amount over both sides of the

5 Atlantic. The comparison of precipitation with TRMM observations reveals that the EMAC model gives more realistic results over West Africa compared with the Caribbean for all convection schemes.

  Second, the ADP (Fig. 11) illustrates the effect of convection schemes on the transatlantic dust transport. Over West Africa, the dust is already aged with ADP values between 0.2 and 0.4, whereas over the Caribbean the ADP values are with 0.3 and 0.5 only slightly higher. The lower ADP values over West Africa can be attributed to the higher dust loadings, which requires

a much larger amount of condensable material to becomes fully aged. Over the Caribbean, the dust loading is considerably lower due to the removal processes along dust transport, which takes about 5 days. This time is sufficiently long for coating by acids and other soluble materials (Gläser et al., 2015), and causes the dust to become more aged. On the other hand, the high precipitation amount at 15°W over the Western Africa region results in higher scavenging of the aged dust particles compared with the –pristine– (non-aged) dust particles. This results in a decrease in the ADP values, in agreement with the results of

Abdelkader et al. (2015). West of 15°W, the dust is transported over the Atlantic into a region where precipitation is much lower (middle panels). Consequently, the level of chemical aging increases. The EMAC reference simulation (with too strong precipitation) therefore shows a higher ADP (0.35 compared to 0.2), which is a result of the lower dust burden and mainly caused by a too efficient wet removal.

  Thus, the convection sensitivity analysis points to a too strong removal mechanism of the mineral dust particles along

transatlantic transport, when the default convection scheme is used in EMAC. In addition, the level of dust chemical aging seems to control the efficiency of dust scavenging. Higher levels of aged dust, and higher precipitation amounts, significantly decrease the dust burden and thus the AOD over the Caribbean. This further suggests that modeling the transatlantic dust transports requires improved convection parameterization (i.e., more realistic precipitation rates), and in parallel a realistic representation of dust chemical aging.

## 25 4.3 Dust chemical aging

To further investigate the impact of the dust chemical aging on the transatlantic dust transport, this process was excluded for an additional sensitivity study. The level of dust chemical aging depends on the availability of condensable acids (see Sec. 2). For the "No Aging" case, the condensation of acids on insoluble dust particles is excluded, which suppresses water uptake by dust particles. Figure 12 shows the AOD time series at the AERONET stations on both sides of the Atlantic for the two cases, i.e.,

Aging and No Aging. Generally, the Aging case systematically shows a higher AOD compared to the No Aging case, which emphasizes the importance of this process and the associated water uptake in agreement with the results of Abdelkader et al. (2015). However, the dust chemical aging has a stronger impact on the AOD over West Africa, especially at the Capo Verde and Dakar stations during the two dust outbreaks discussed above. The Aging case shows about 0.2 higher AOD compared with the No Aging case as a result of the larger particle size and the associated water uptake. This increases the scattering cross

section and thus the AOD. Over the Caribbean, the dust chemical aging shows a smaller impact on the AOD; the Aging case shows only about 0.05 higher AOD because of the lower contribution of the dust to the overall AOD values (which includes the contribution of other aerosol species, sea salt, etc., for instance). During the high dust outbreaks, the concentration of the soluble compounds required to coat such a large amount of dust is not available according to the EMAC model. The aged dust

particles are removed more efficiently during transport and relatively more uncoated dust particles reach the Caribbean. As a result, the dust chemical aging has a limited effect on the AOD over the Caribbean AERONET stations.

    Figure 13 shows the regional difference (monthly mean) for (a) the dust burden, (b) AOD, (c) dust emissions averaged over the region from $18°$-$22°$N, and (d) the dust-only AOD (No Aging minus Aging case). The results show a higher dust burden over the source regions in West Africa for the No Aging case compared with the reference (Aging). For the No Aging case,

the dust plume slightly extends further to the west over the Caribbean because of the reduced dust removal during transport. The difference between the two simulations decreases during transport, which is supported by the differences in the dust-only AOD. In contrast, the difference in the total AOD shows lower values over the dust source region compared with the Aging case, which indicates a significant contribution of the dust chemical aging to the total AOD.

    Interestingly, the negative feedback between the AOD and the radiation scheme results in higher dust emission over the

region from $10°$E to $0°$and thus causes a higher dust burden. The average dust emission during July 2009 over the region from $18°$N to $22°$N (lower panel) shows that the dust emission for the "Aging" case is on average higher by about $3\,\mathrm{g\,m^{-2}}$, which results in a higher dust burden by $1\,\mathrm{g\,m^{-2}}$ while the remaining amount of the dust ($2\,\mathrm{g\,m^{-3}}$) is deposited. The higher AOD in the "Aging" case results in stronger scattering of short-wave solar radiation, lower surface radiation fluxes but higher surface wind speed (as shown in Fig. S7 in the supplement), and eventually stronger dust emission of $2\,\mathrm{g\,m^{-2}}$. The increased wind

speed (more than $0.25\,\mathrm{m\,s^{-1}}$ on monthly average) could result either from the increase in the surface temperature because of the absorption of the dust particles and the resultant increase in the surface pressure (Menon, 2002; Mishra et al., 2014) or from a change in the horizontal temperature gradient that also increases the local wind speed (Rémy et al., 2015). On the other hand, the more efficient removal of the large dust particles in the Aging case by both scavenging and sedimentation results in lower dust burden and thus the lower AOD. The balance between the two competing processes defines the impact of dust chemical

aging on AOD. The difference in the dust-only optical depth is shown in the lower right panel of Fig. 13 and indicates that the "No Aging" case has higher dust optical depth as a result of the lower dust removal as compared with the "Aging" case. The difference is at a minimum within a region between $18°$N to $22°$N. However, the total AOD shows that the "No Aging" case leads to a lower AOD, which is significant over Western Africa and less pronounced over the Caribbean sites. Note that the AOD, as compared with AERONET stations, shown in Fig. 12 does not resolve this large difference because the AERONET

stations are all located in the DTA region where the differences are obviously lower.

    The substantial higher AOD for the "Aging" case (0.3 on monthly mean) primarily results from the dust chemical aging because of the associated water uptake. Figure 14 shows the monthly averaged burden for lumped gas-phase acids (sum of $HCl+HNO_3+H_2SO_4$) and the difference between both simulations. The figure also shows the corresponding lumped inorganic aerosol mass (sum of $SO_4^{2-} + HSO_4^- + NO_3^- + NH_4^+ + Cl^- + Na^+ + Ca^{2+} + K^+ + Mg^{2+}$) and the aerosol associated water mass.

For the Aging case, the burden of acids is very low over the dust source region because of the uptake by dust particles –

an important effect which has been also recently studied with the EMAC model by Karydis et al. (2016) for the nitric acid uptake (also included here). Consequently, the aerosol burden is higher over the dust source region and over the outflow region, because of the additional neutralization of the calcium ions by anions and the associated absorption of water vapor by the resulting calcium salts. As a result, the aerosol-associated water increases by more than $255\,\mathrm{mg\,m^{-2}}$ for the Aged case. The effect of dust chemical aging is a result of gas–aerosol partitioning that clearly affects the AOD. It is best observed in the differences (right column of Figure 14), which reveal that the impact of dust chemical aging can be significant, but mainly due to the associated uptake of aerosol water. We refer to this effect as the "direct effect of dust chemical aging." In addition, we refer to the higher removal of aged dust (by both sedimentation and scavenging), and the consequently shorter dust lifetime, as the "indirect effect of dust chemical aging" – both effects are introduced in this study.

To obtain improved statistics for the effect of dust chemical aging, the same analysis (Aging versus No Aging) was applied to the entire evaluation period (2000–2012) at lower T42 model resolution. Figure 15 shows the long-term meridional dust burden mean and the model precipitation for TRMM observations over the DTA and DIR zones (as discussed above). The No Aging case consistently shows higher dust burdens in the DIR zone as a result of more efficient scavenging for the Aging case. Even for this long-term average, the dust burden is three times higher for the "No Aging" case than the "Aging" case over the Caribbean sites. However, the impact of scavenging in the Aging case is stronger in the region between $10°$W and $20°$W, which corresponds with the precipitation peak in the West Africa region.

## 5 Conclusions

Tansatlantic dust transport is a major large-scale atmospheric phenomenon. Although the EMAC model mostly reproduces the dust pattern during the transatlantic dust transport, the dust loadings and AOD can deviate in magnitude and seasonality from observations. To examine the controlling processes, the dust outflow region has been divided into two subregions: (1) the dust-ITCZ (DIZ) zone and (2) the adjacent dust transport over the Atlantic Ocean (DTA) zone. In the former, the dust is removed primarily by scavenging, whereas in the latter region sedimentation is predominant. Considering the two subregions allows the distinction of factors that affect the transatlantic dust transport.

Several sensitivity studies were conducted using the EMAC model following Abdelkader et al. (2015) – with a comprehensive setup which includes a fully coupled online dust emission scheme and an explicit chemical aging of the atmospheric dust particles. First, the simulated AOD is sensitive to the emission flux parameterization, and even more to the choice of the convection scheme. The dust emission flux affects the AOD over West Africa more strongly compared to the Caribbean sites. On the other hand, the dust burden shows a very steep gradient westward over the Atlantic Ocean. This is mainly a result of dust removal by deposition (sedimentation and scavenging) during long-range transport. Over the Atlantic (within DTA), this gradient is linear in the logarithmic scale, whereas the gradient is nonlinear over the Western and Eastern Atlantic (especially within DIZ). The dust burden over West Africa (east of $10°$W) is about $1000\,\mathrm{\mu g\,m^{-3}}$ but declines to $50\,\mathrm{\mu g\,m^{-3}}$ over the Caribbean. The different convection parameterization schemes show more than a factor of 2 difference in dust burden over West Africa, and about a factor of 3 over the Caribbean. This is primarily a result of different precipitation rates and the associated differ-

ences in dust removal. Overall, the dust outflow into the Caribbean is best represented by the ECMWF convection scheme, as a result of more realistic representation of precipitation within the ITCZ (compared to other schemes available in EMAC and relative to TRMM observations). The more realistic precipitation subsequently improves the dust removal (compared to the reference EMAC simulations) and subsequently the AOD on both sides of the Atlantic Ocean significantly within the DIZ zone, a region which is largely controlled by wet removal processes. Considering the dust chemical aging amplifies this effect.

To study the impact of dust chemical aging, we use the calcium cation as a proxy for the overall chemical reactive dust fraction and consider the uptake of major inorganic acids (i.e., $H_2SO_4$, $HNO_3$, $HCl$) and their anions, i.e., sulfate ($SO_4^{2-}$), bi-sulfate ($HSO_4^-$), nitrate ($NO_3^-$), and chloride ($Cl^-$)) on the surface of mineral particles. The subsequent neutralization reactions with the calcium cation forms various salt compounds that causes the uptake of water vapour from the atmosphere, which leads to the chemical aging of dust particles. Dust chemical aging changes the particle sizes because of the additional amount of condensed inorganic acids and the associated uptake of water vapor by the neutralization products (salts). Therefore, the aged dust particles are larger and scatter light more efficiently, whereas they are more rapidly removed by dry and wet removal processes. To analyze these effects, we performed Aging and No Aging simulations, for which we distinguish between the direct and indirect effect of dust chemical aging on AOD.

In our senitivity simulations, the dust chemical aging shows the largest impact on the AOD over West Africa and on the dust burden in the ITCZ. The larger impact on the AOD results from the increase in the aerosol burden (more than $120\,\mathrm{mg\,m^{-2}}$) due to the uptake of acids and associated water by the originally insoluble dust particles. This directly increases the AOD by 0.15 (monthly average). As a result of the radiative feedback on the atmospheric dynamics and circulation, the dust emission regionally increases. On the other hand, the aged dust particles are more efficiently removed in our EMAC reference setup compared with the non-aged dust particles case. The enhanced removal of aged particles decreases the dust burden and lifetime, indirectly affecting the AOD. Both processes are significant and the net effect depends on the region and the level of dust chemical aging, which is controlled by the strength of the dust outflow and the collocated air-pollution levels. In order to improve the dust cycle in climate models, we recommend an explicit treatment of dust chemical aging, at least by considering the calcium cation as a proxy for the overall chemical reactivity of the mineral dust particles.

## Appendix A: Evaluation metrics

– RMSE – Root Mean Square Error between the model (m) and the observations (o):

$$RMSE = \sqrt{\frac{1}{N}\sum(X_m - X_o)^2} \tag{A1}$$

– $\sigma$ – Standard deviation of the model ($\sigma_m$) and the observation ($\sigma_o$) for variable ($X_i$) with average of ($\bar{X}$) with N the number of observations:

$$\sigma = \sqrt{\frac{1}{N}\sum_{i=1}^{N}(X_i - \bar{X})^2}, \qquad where \quad \bar{X} = \frac{1}{N}\sum_{i=1}^{N}X_i \tag{A2}$$

- R – Correlation coefficient between the model (m) and the observations (o):

$$R = \frac{\sum_{i=1}^{N}(X_i^m - \bar{X^m})(X_i^o - \bar{X^o})}{\sum_{i=1}^{N}(X_i^m - \bar{X^m})^2 \sum_{i=1}^{N}(X_i^o - \bar{X^o})^2} \tag{A3}$$

- r geometric mean of the model ($r_m$) and the observations ($r_o$).

$$r = \sqrt[n]{\Pi_{i=1}^{N} X} \tag{A4}$$

- MBE – Mean Bias Error between the model and the observations:

$$MBE = \frac{1}{N}\sum(X_m - X_o) \tag{A5}$$

- GFE Growth Factorial Error

$$GFE = \frac{1}{N}\sum \frac{|(X_m - X_o)|}{X_m + X_o} \tag{A6}$$

- SS1 – Skill score between the model (m) and the observations (o) (Taylor, 2001):

$$SS1 = \frac{4(1+R)}{(\sigma_f + 1/\sigma_f)^2(1+R_0)}, \qquad where \quad \sigma_f = \frac{\sigma_o}{\sigma_m} \quad R_0 = 0.0 \tag{A7}$$

*Acknowledgements.* All simulations in this study were carried out on the Cy-Tera Cluster. The Project Cy-Tera ( NEA-Y Π O Δ OMH/ Σ TPATH/0308/31) is co-financed by the European Regional Development Fund and the Republic of Cyprus through the Research Promotion Foundation. This study has received funding from the European Research Council under the European Union's Seventh Framework Program (FP7/2007-2013) / ERC grant agreement no 226144. The authors are grateful to NASA and PHOTONS (PHOtomEtrie pour le Traitement OpErationnel de Normalisation Satellitaire; Univ. of Lille 1, CNES, and CNRS-INSU), a federation of ground-based remote sensing aerosol networks who establish the AErosol RObotic NETwork program (http://aeronet.gsfc.nasa.gov), for the AERONET data used in this study. G.S. is supported by King Abdullah University of Science and Technology (KAUST) CRG3 grant.

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

**Table 1.** EMAC sub-models used in the current study and the corresponding references.

| Submodel | Description | Reference |
|---|---|---|
| AEROPT | Aerosol optical properties | Lauer et al. (2007); Klingmüller et al. (2014); Pozzer et al. (2015) |
| CLOUD | ECHAM5 cloud scheme as MESSy submodel | Roeckner et al. (2006) |
| CONVECT | Convection parameterizations | Tost et al. (2010) |
| CVTRANS | Convective tracer transport | Tost et al. (2006b) |
| DDEP | Dry deposition | Kerkweg et al. (2006a) |
| EQSAM4clim | Fast aerosol thermodynamics | Metzger et al. (2016) |
| GMXe | Aerosol dynamics and microphysics | Pringle et al. (2010) |
| ISORROPIA-II | Aerosol thermodynamics | Fountoukis and Nenes (2007) |
| JVAL | On-line photolysis rates | Landgraf and Crutzen (1998) |
| LNOX | $NO_x$ production from lightning | Tost et al. (2007) |
| MECCA | Gas phase chemistry | Sander et al. (2005) |
| OFFEMIS | Prescribed emissions of trace gases and aerosols | Kerkweg et al. (2006a) |
| ONEMIS | On-line calculated emissions | Kerkweg et al. (2006b); Astitha et al. (2012) |
| RAD | ECHAM5 radiation scheme as MESSy submodel | Roeckner et al. (2006); Joeckel et al. (2010) |
| SCAV | Comprehensive scavenging of aerosols and gases | Tost et al. (2006a) |
| SEDI | Sedimentation of aerosols | Kerkweg et al. (2006a) |
| TNUDGE | Newtonian relaxation of species | Kerkweg et al. (2006a) |
| TROPOP | Tropopause and other diagnostics | Joeckel et al. (2006) |

**Table 2.** Long-term EMAC model evaluation for the period 2000-2012 for AOD. Statistics are given for both sides of the Atlantic, based on the selected AERONET sites around West Africa and the Caribbean (station average). The evaluation metrics are defined in Appendix A, while the station locations are shown in Figure 3.

|  | Western Africa | Caribbean |
|---|---|---|
| $\text{Mean}_m$ | $0.16\pm 0.27$ | $0.12\pm 0.18$ |
| $\text{Mean}_o$ | $0.24\pm 0.37$ | $0.14\pm 0.22$ |
| $r_m$ | $0.13\pm 0.40$ | $0.11\pm 0.27$ |
| $r_o$ | $0.29\pm 0.35$ | $0.13\pm 0.29$ |
| RMSE | 0.35 | 0.23 |
| R | 0.61 | 0.43 |
| MBE | -0.19 | -0.11 |
| GFE | -0.24 | -0.12 |
| SS1 | 0.73 | 0.70 |
| PF2 | 0.59 | 0.81 |
| PF10 | 1.00 | 1.00 |
| NPOINTS | 50288 | 15827 |

**Table 3.** Description of the transatlantic dust transport sensitivity simulations for two key-processes: (i) Emission flux (Sec. 4.1) and (ii) convection scheme (Sec. 4.2). Highlighted cases are shown in the manuscript (for all cases see the Supplement, Fig. S3–S4). The emitted dust mass during July 2009 for the reference case is $0.6133 \, \mathrm{kg \, m^{-2}}$.

|  | Case | Description |
|---|---|---|
|  | EMAC | Reference simulation |
|  | B1E1 | Redistribution of dust between accumulation and coarse modes |
|  | B1E2 | As EMAC, accumulation fraction incased by a factor of 2.61 |
|  | B1E3 | As EMAC, the coarse mode increased by a factor of 5.3 |
| Emission | B1E4 | As EMAC, the accumulation mode increased by a factor of 5.3 |
|  | B1E5 | As EMAC, the accumulation mode increased by a factor of 10.6 |
|  | B1E6 | As EMAC, the accumulation and coarse modes increased by a factor of 10.6 and 2.61 respectively |
|  | B1E7 | As EMAC, the accumulation and the coarse modes increased by a factor of 2.61 |
|  | B1E8 | As EMAC, factor=2.61 in the horizontal flux |
|  | EMAC | Reference simulation; TIEDTKE convection with NORDENG closure |
|  | B1T2 | TIEDTKE convection with TIEDTKE closure (Tiedtke, 1989) |
| Convection | B1T3 | TIEDTKE convection with HYBRID closure (Tiedtke, 1989) |
|  | B1T4 | ECMWF operational convection scheme (Bechtold et al., 2004) with the shallow convection closure of Grant and Brown (1999) |
|  | B1T5 | ECMWF operational convection scheme (Bechtold et al., 2004) |
|  | B1T6 | Zhang-Hack-McFarlane convection scheme (Zhang and McFarlane, 1995; Hack, 1994) |

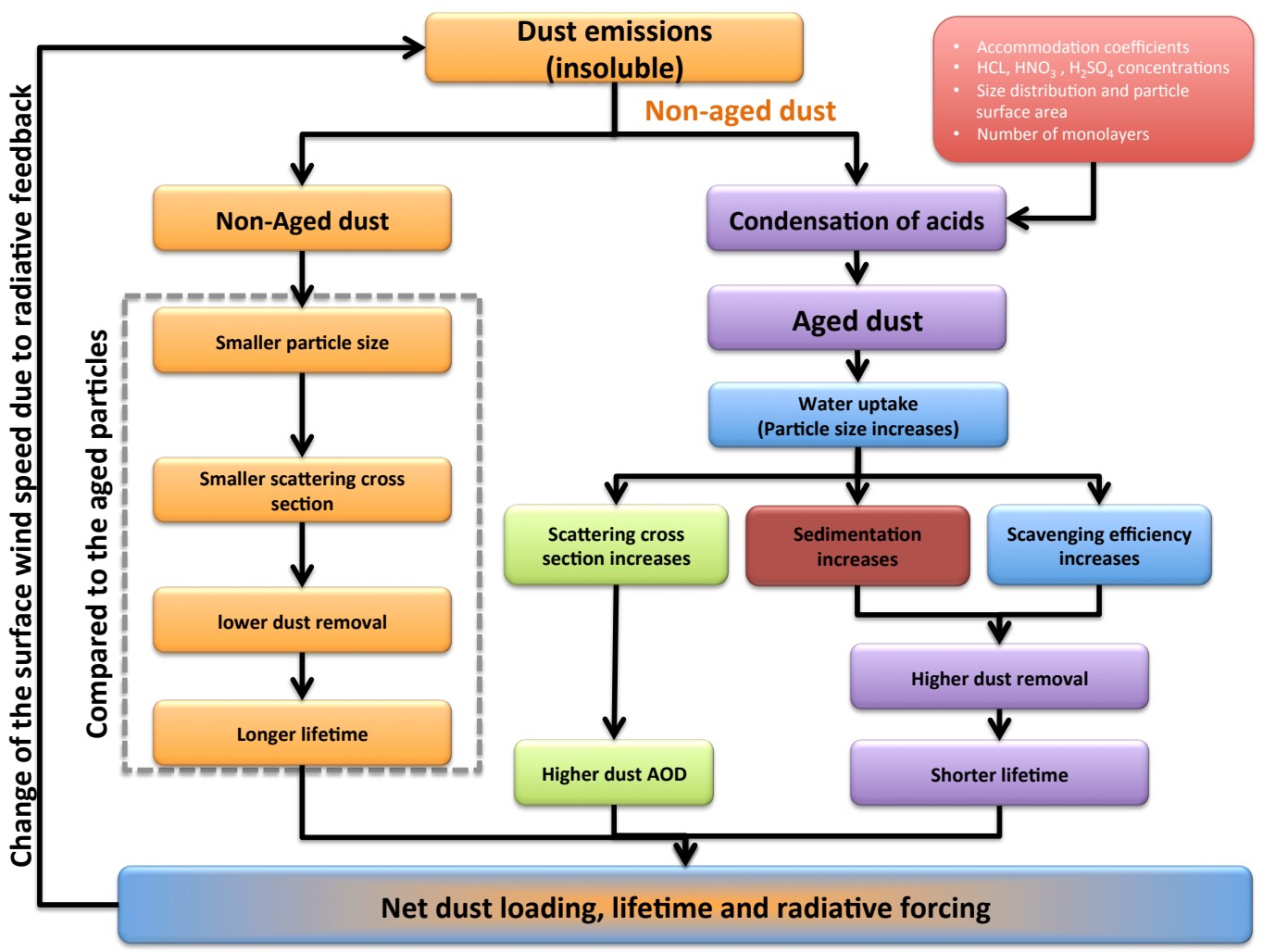

**Figure 1.** Schematic representation of the dust cycle and air-pollution-dust-chemical-aging-radiation feedbacks in EMAC. Air pollution controls the chemical aging of dust particles, whereby the consequent water uptake increases the dust particle scattering cross section, enhances the dust deposition (wet and dry) which tends to decrease the dust lifetime. The net radiative differences between aged and non-aged dust particles are indicated.

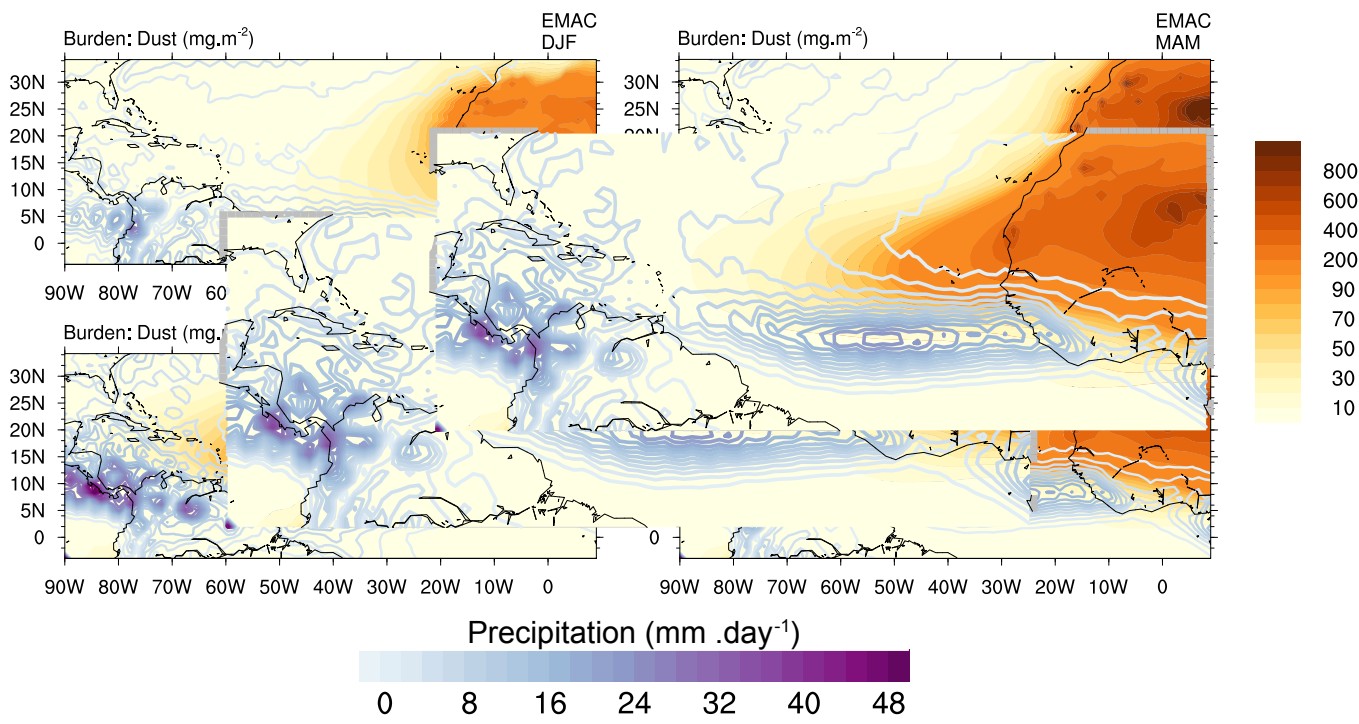

**Figure 2.** Seasonal averages of the dust burden and precipitation representing the transatlantic dust outflow for the entire model evaluation period (2000-2012). Dust burden and precipitation are at maximum during boreal summer and at minimum during winter. The orange color represents the dust burden while the purple color (contour lines) depicts precipitation.

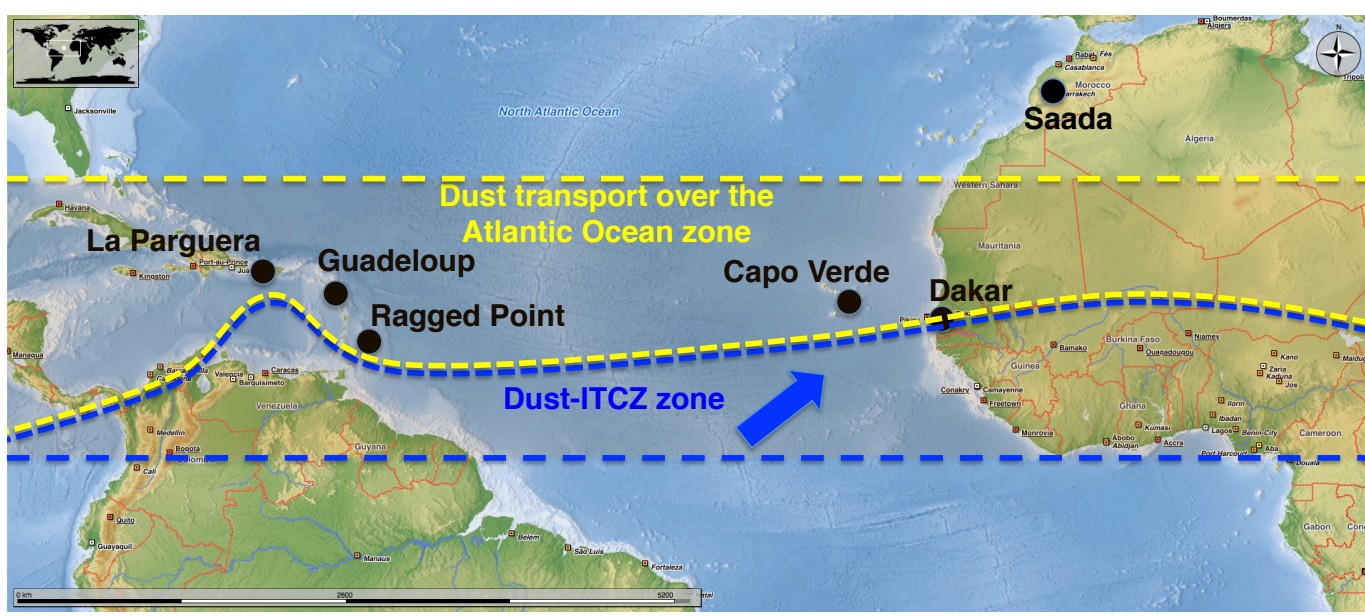

**Figure 3.** The location of selected AERONET stations used in the transatlantic dust transport study. Stations: Saada, Capo Verde and Dakar are as *"West Africa"* and stations: La Parguera, Guadeloup and Ragged Point as *"Caribbean"*. The upper blue line shows the approximate northern bound of the ITCZ, The yellow box indicates the adjacent dust transport region (DTA) zone. The region within the blue bounds represents the dust-ITCZ interaction zone (DIZ). These regions are defined according to the predominance of the dust removal mechanism shown in Fig. S1 in the supplement.

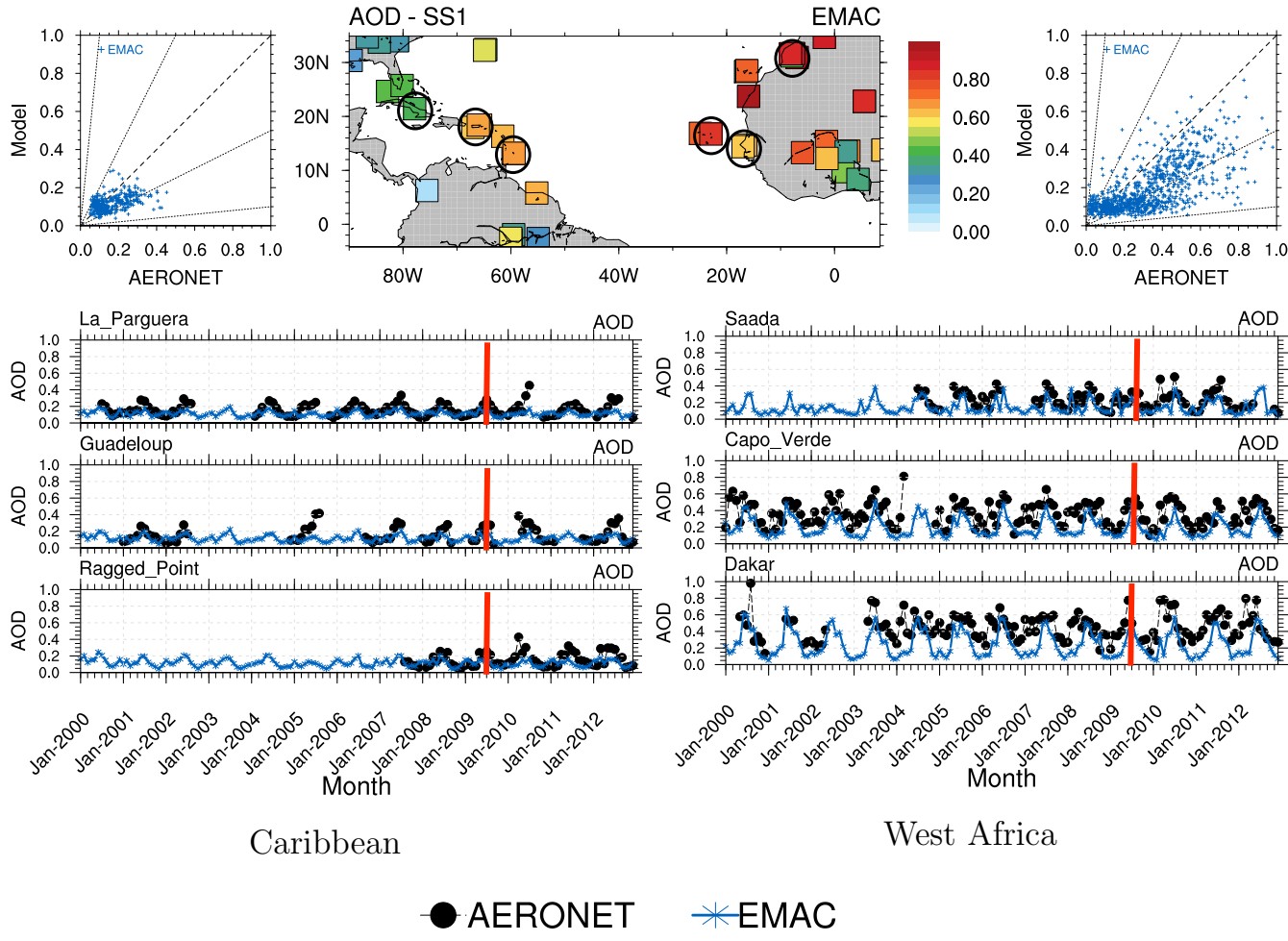

**Figure 4.** Long-term evaluation for AOD (2000-2012) over western Africa and the Caribbean: (Top panel) scatter plots including all observations from all stations (left for the Caribbean, right for the West Africa region) and skill score (SS1) defined in Appendix (A): (Lower panel) time series for stations at both regions (monthly means of 5 hour averages for model and AERONET AOD). The red bars represent the July 2009 dust outflow period and the black circles depict the selected AERONET stations shown in Fig. 3 within observations for the period of our sensitivity simulations. The dotted lines in the scatter plot show the 1:2 and 1:10 ratio.

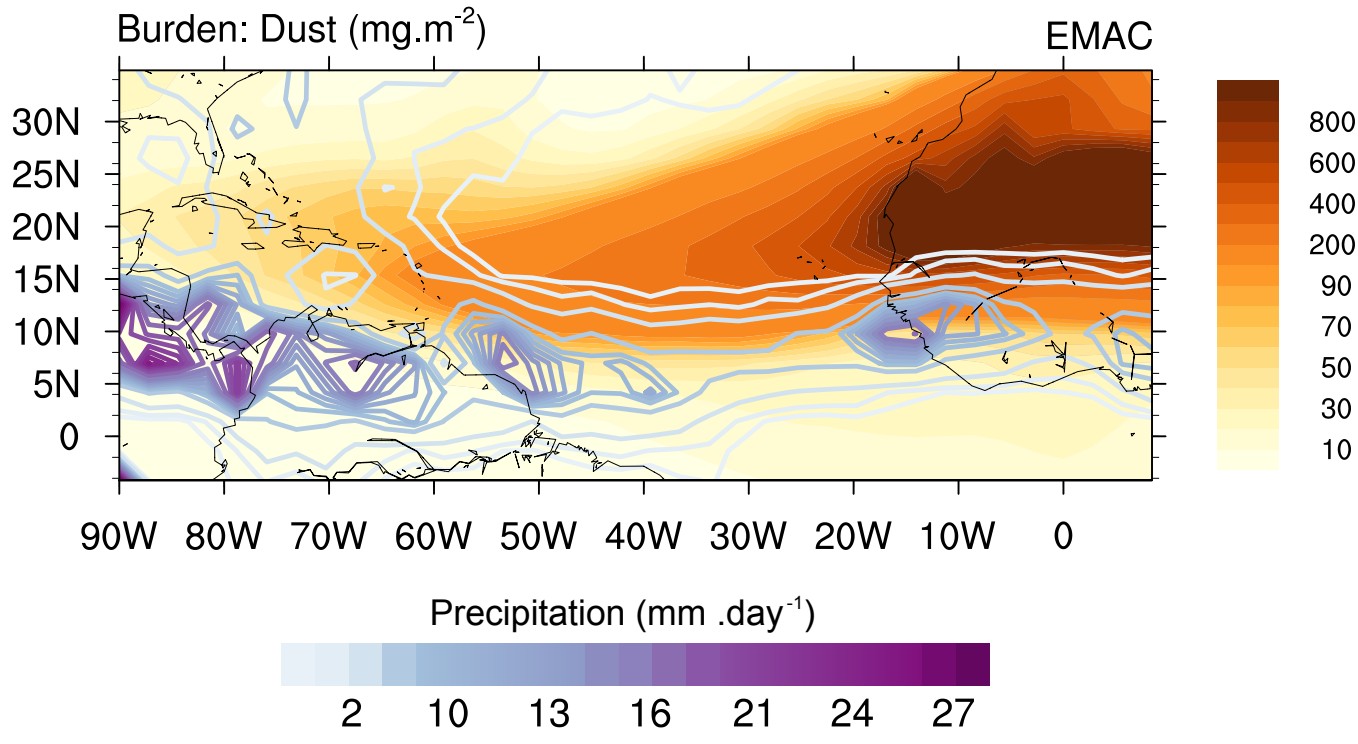

**Figure 5.** The EMAC computed spatial distribution of the dust burden (orange) and total precipitation (purple lines) for the reference simulation for July 2009 (monthly mean).

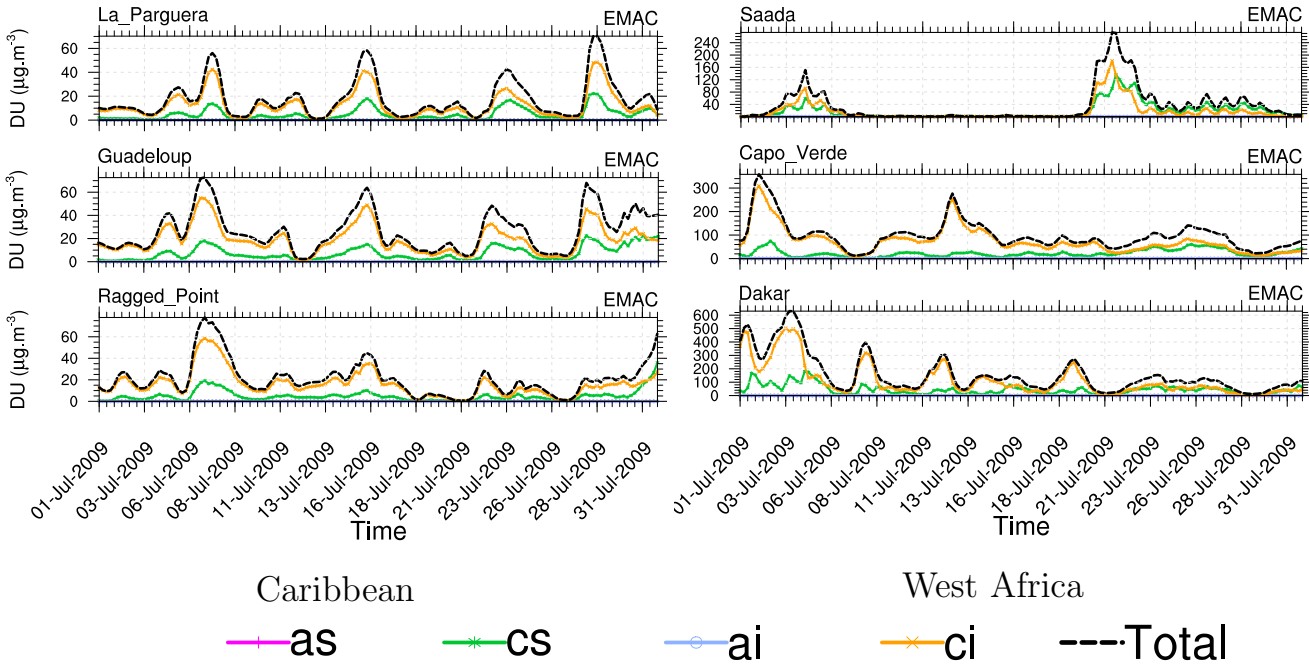

**Figure 6.** Time series of size-resolved surface dust concentrations for the different AERONET stations shown in Figure 3. Aerosol modes: accumulation soluble (as); coarse soluble (cs); accumulation insoluble (ai); coarse insoluble (ci). Note the different scaling which reflects the wide range of concentrations at these stations. The accumulation mode dust fraction has a much lower contribution to the total dust concentration.

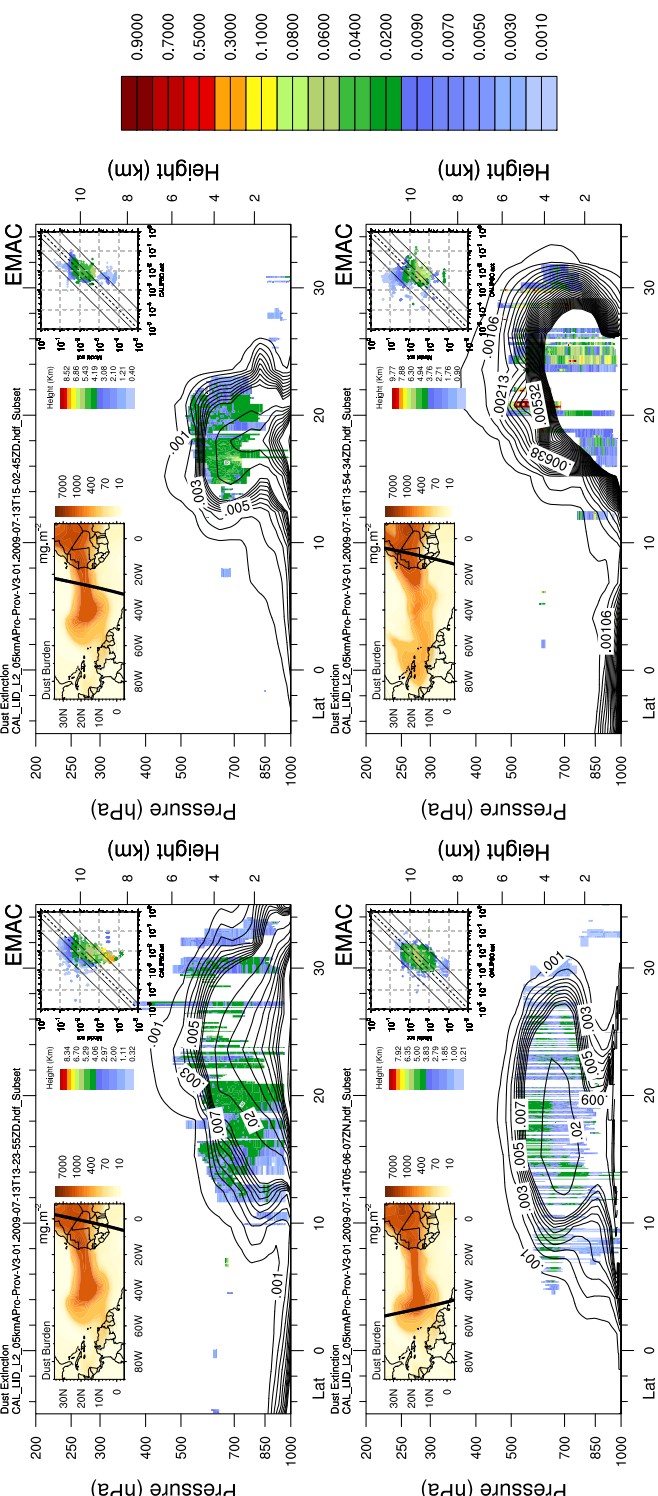

**Figure 7.** Collocated EMAC and CALIPSO observations of dust extinction and burden for four different CALIPSO overpasses during the second dust outbreak over the Atlantic Ocean. The time of the overpass is shown in the upper left corners (13-16th July 2009). The solid lines show the simulated extinction and the colored contours shows the observed CALIPSO extinction, which are complemented by the scatter plots for point-to-point comparison colored by the corresponding elevations of each observation (distinguished by the colors). The lines in the scatter plots delineate the one-by-one, the factor two and the factor of ten intervals. All available comparisons with CALIPSO overpasses for this period are shown in the Supplement (Fig. S2a – Fig. S2e).

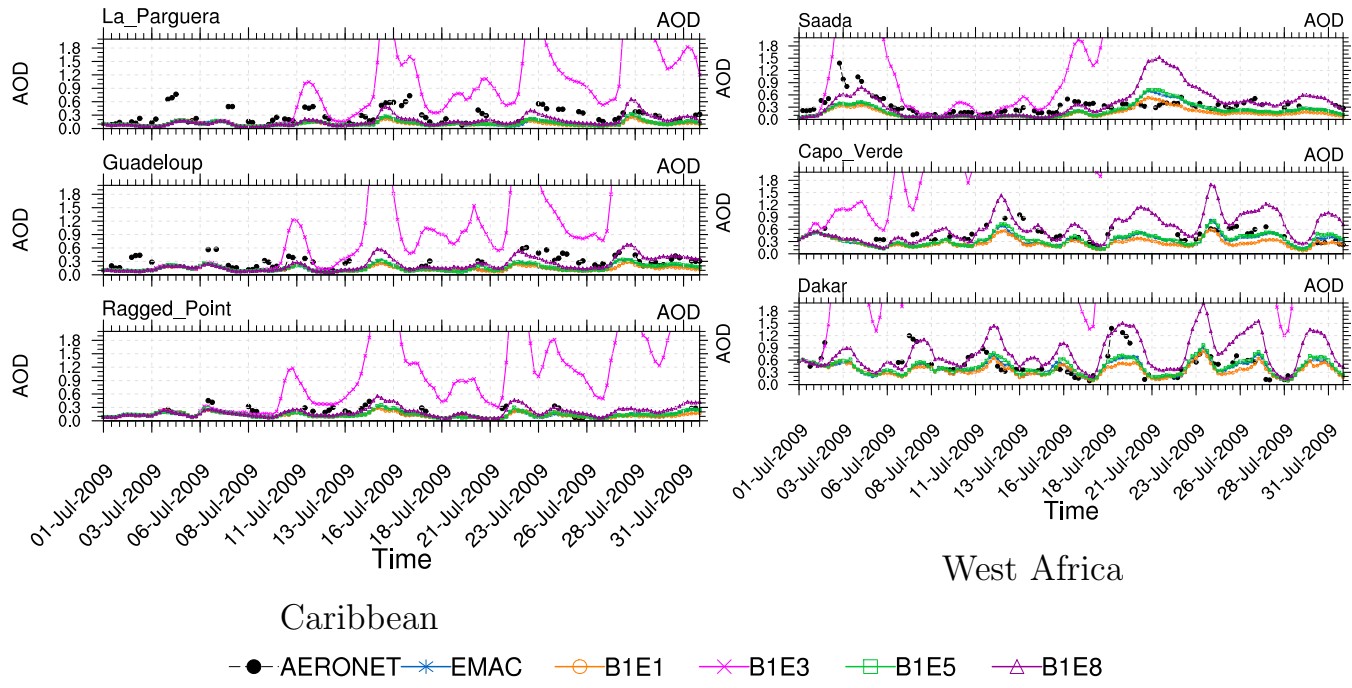

**Figure 8.** EMAC and AERONET AOD for the western Africa (right) and Caribbean (left) sites based on different dust emissions (Table 3).

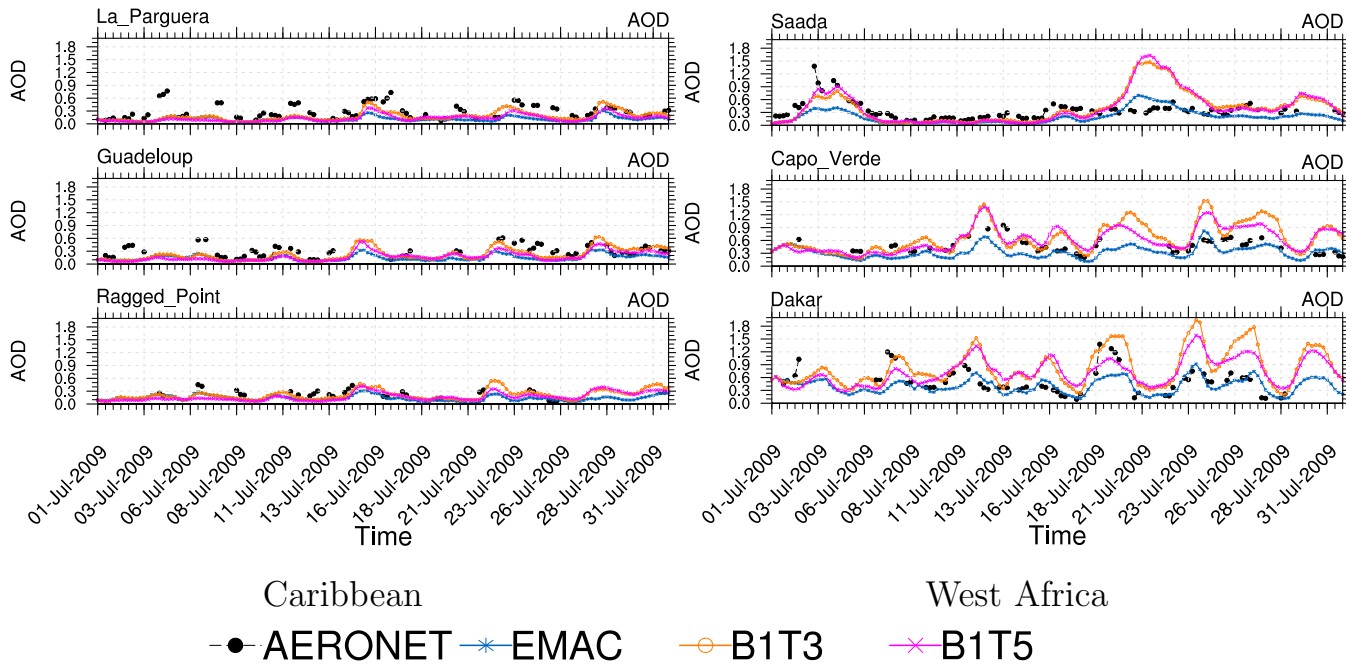

**Figure 9.** EMAC and AERONET AOD for West Africa (right) and Caribbean (left) based on different convection schemes (Table 3).

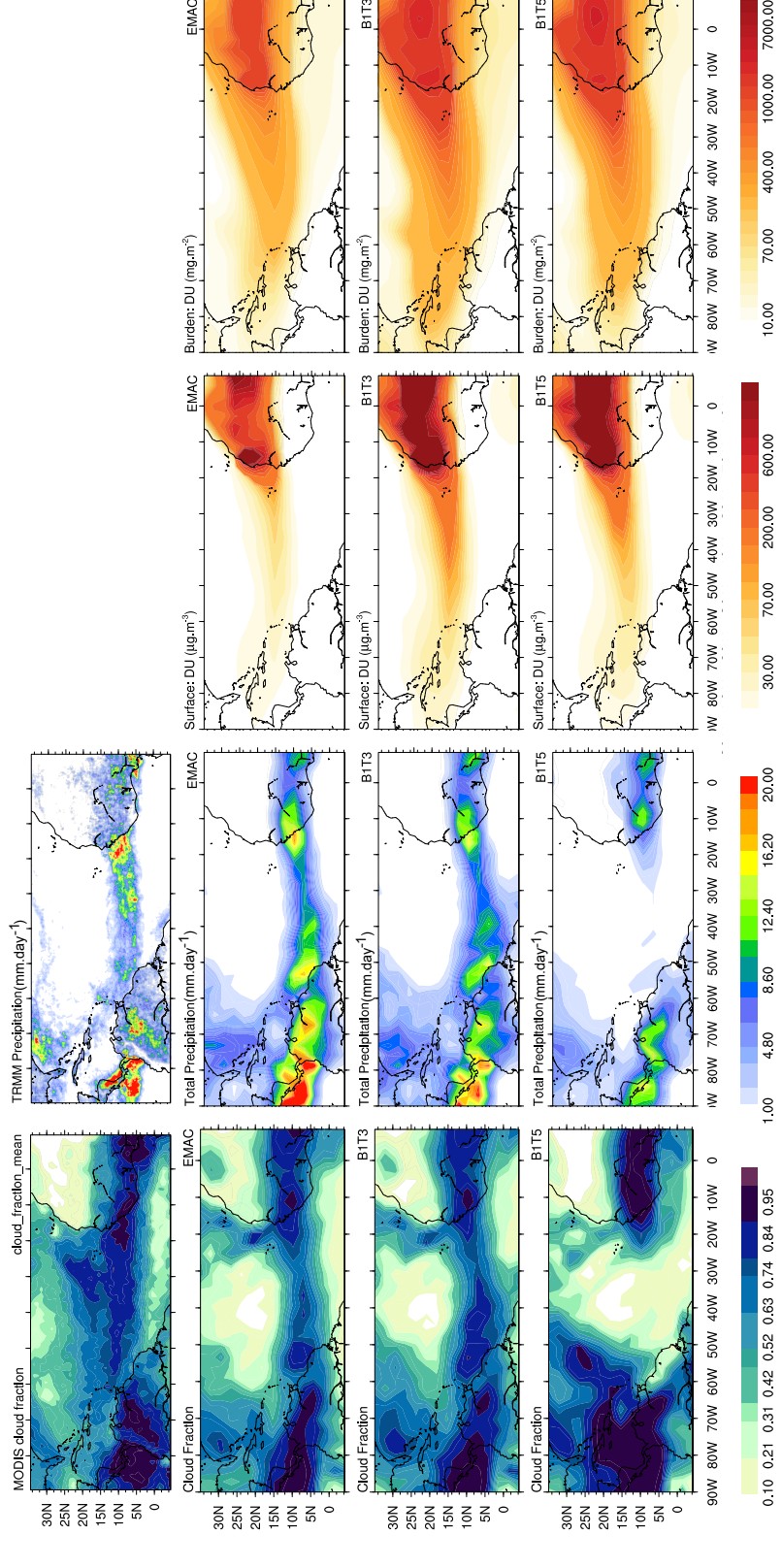

**Figure 10.** (Top) MODIS cloud fraction and TRMM precipitation (July 2009 monthly mean); (below) EMAC results (from left to right) cloud fraction, precipitation, surface dust concentration, and dust burden for different convection schemes (2nd–4th row) highlighted in Table 3. The model precipitation and cloud cover agrees for our EMAC set-up best with TRMM and MODIS observations with the TIEDTKE (B1T3) and ECMWF (B1T5) convection schemes.

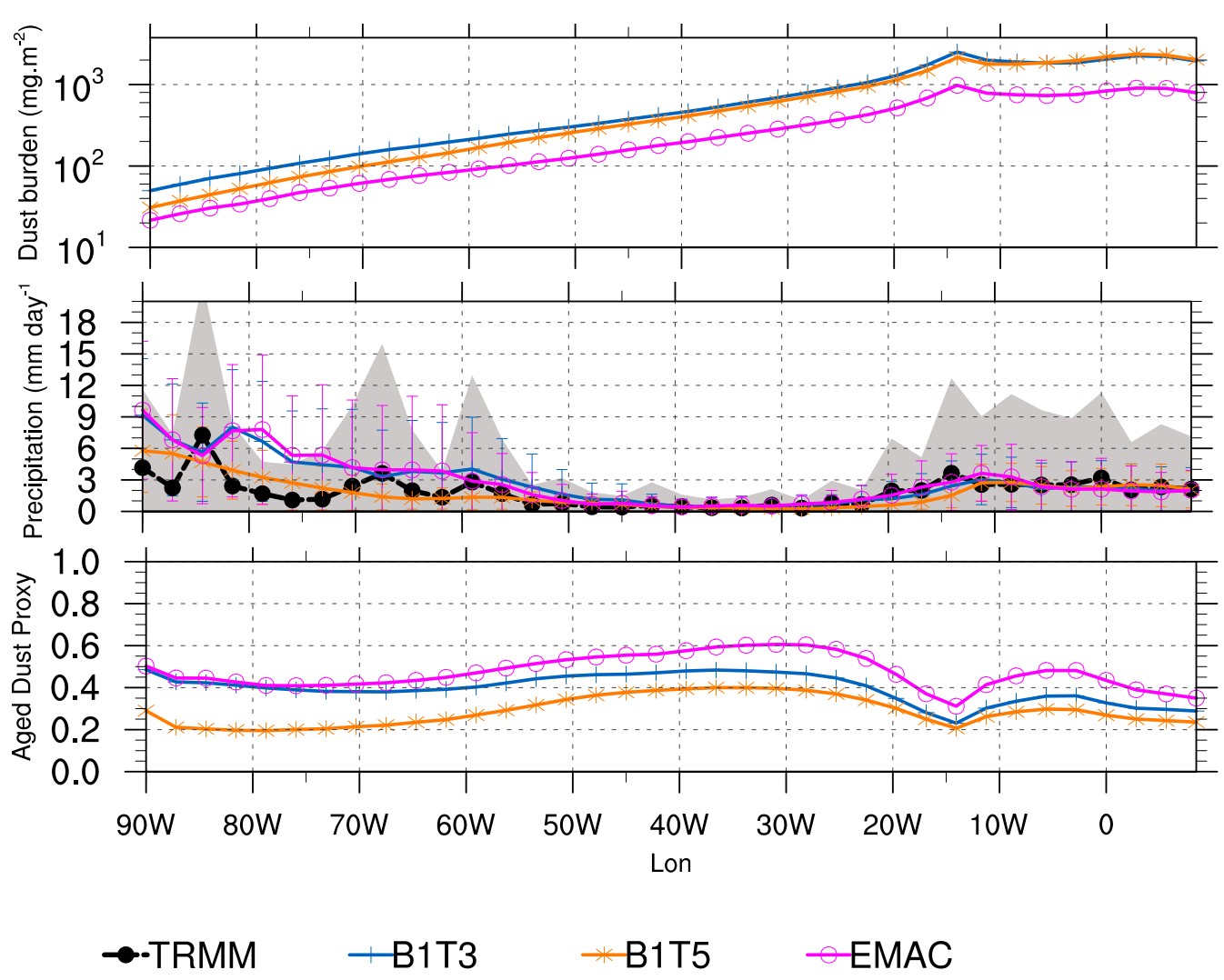

**Figure 11.** Comparison of observed and calculated meridional means over the dust outflow over the Atlantic Ocean region (10°–25°N) for: (top) dust burden, (middle) precipitation, (bottom) aged dust proxy (ADP) for July 2009 (monthly mean). The ADP represents the ratio between aged and non-aged dust particles. The shaded area represents one standard deviation of the TRMM-precipitation and the bars shows one standard deviation of the model results.

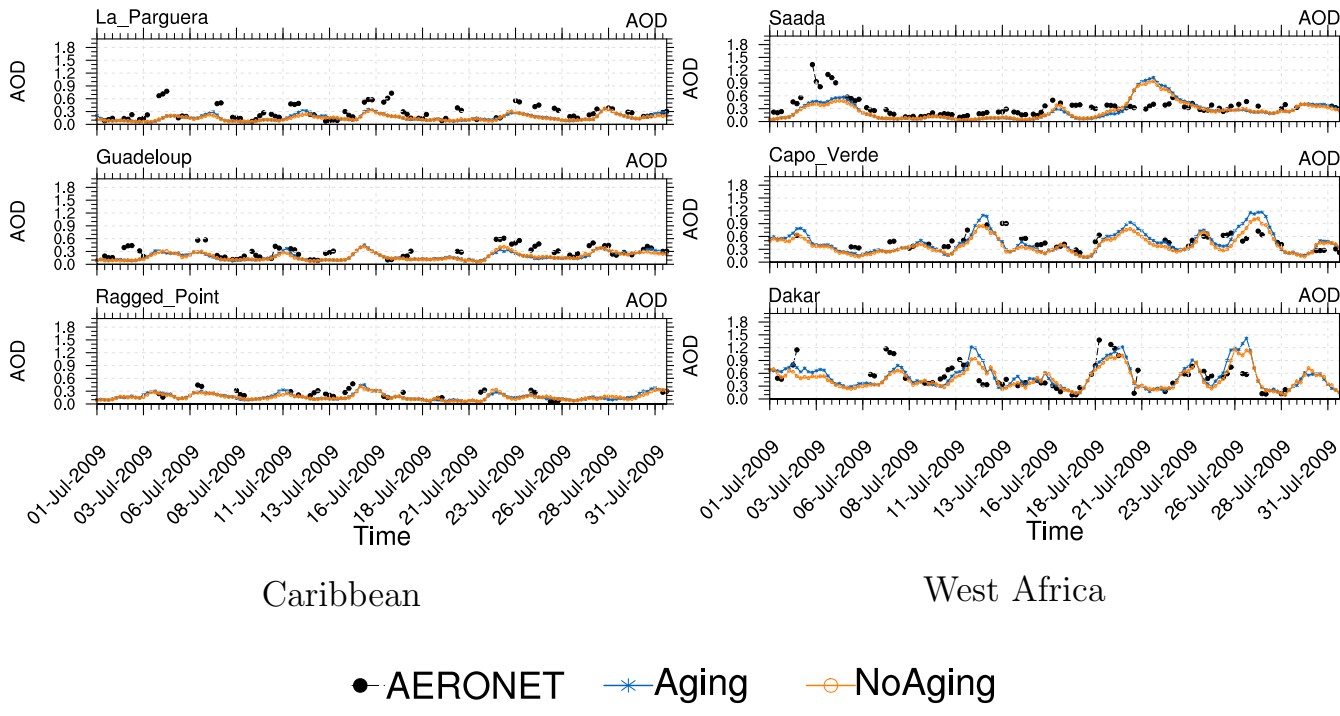

**Figure 12.** Comparison of observed (AERONET) and calculated AOD for western African and the Caribbean and for two EMAC simulations that include and exclude chemical aging (labeled "Aging" and "No aging", respectively).

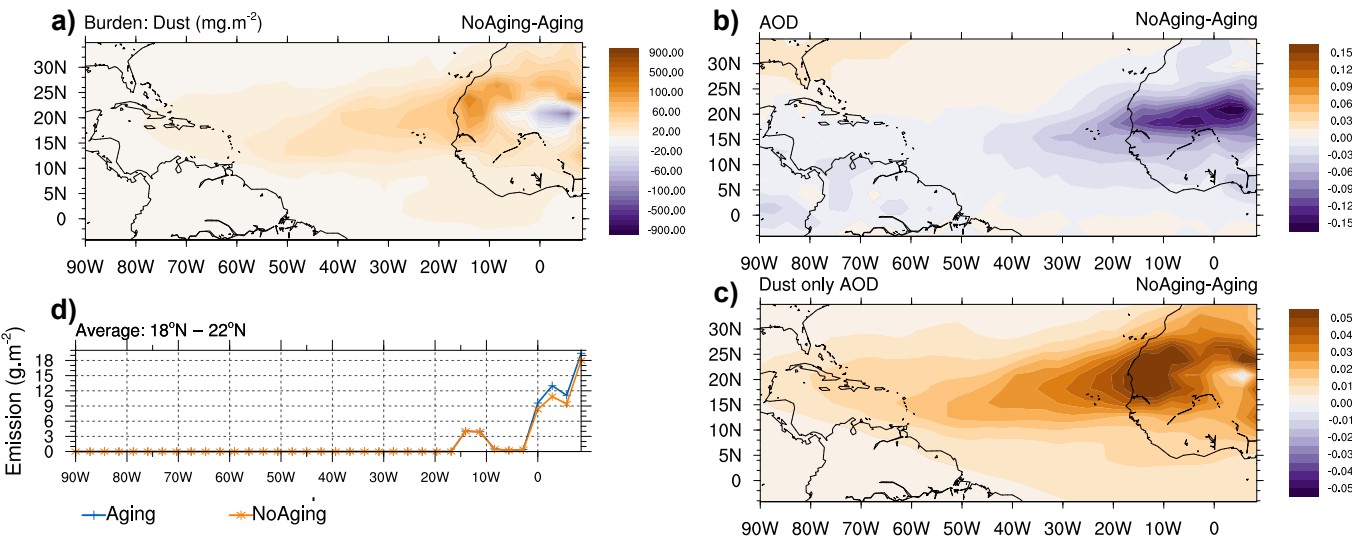

**Figure 13.** EMAC results (monthly mean for July 2009) for two simulations that include and exclude chemical aging (labeled "Aging" and "No aging", respectively). (a) difference in dust burden, (b) difference in AOD, (c) dust emission averaged over the region from 18°-22°N for both simulations, (d) difference in "dust only AOD". "Aging" is the reference case. The difference shows the results of the "No Aging" minus "Aging" case.

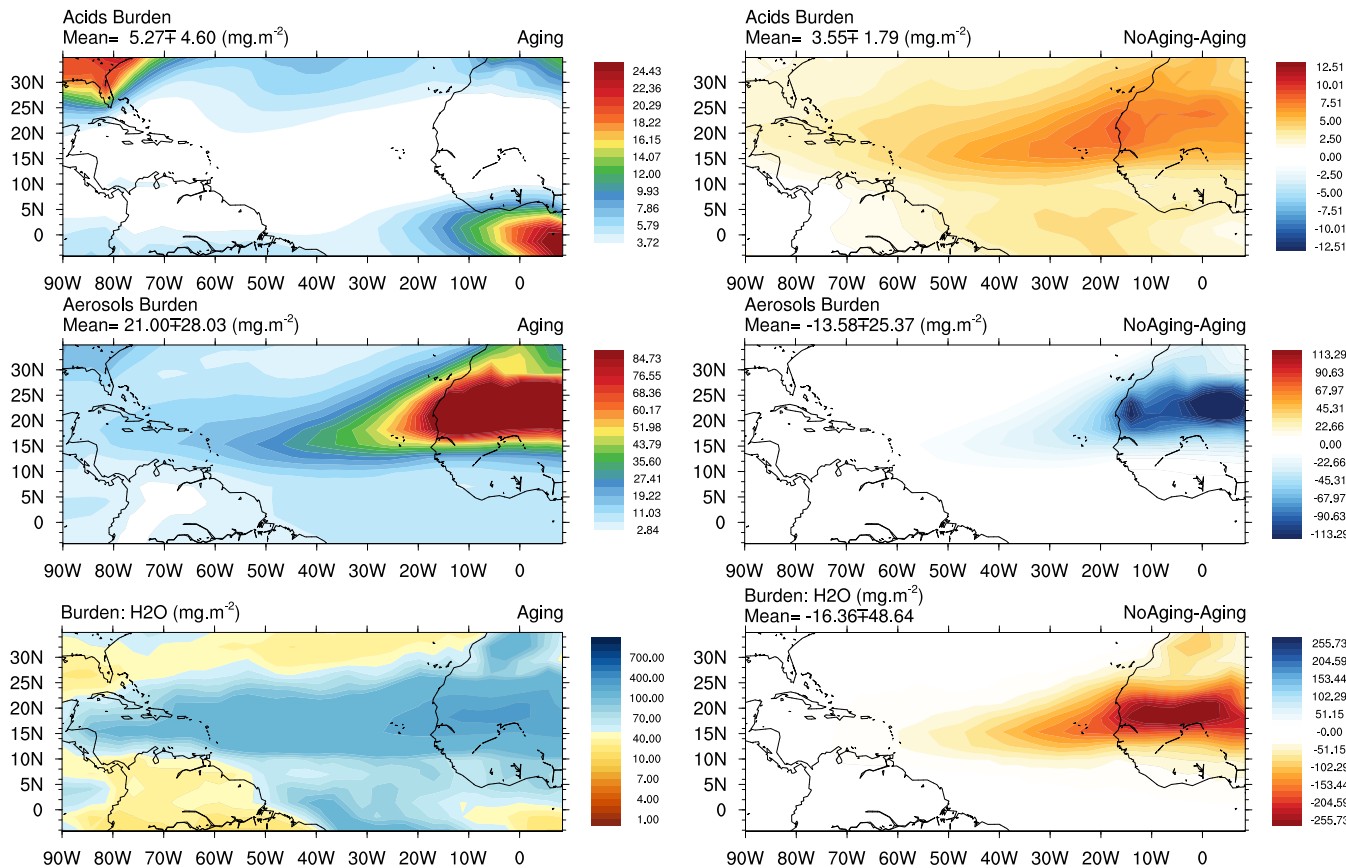

**Figure 14.** Monthly mean (July 2009) for: (top) burden of lumped inorganic gas-phase acids (sum of $HCl+HNO_3+H_2SO_4$), (middle) burden of lumped aerosols (sum of $SO_4^{2-} + HSO_4^- + NO_3^- + NH_4^+ + Cl^- + Na^+ + Ca^{2+} + K^+ + Mg^{2+}$), (bottom) burden of aerosol associated water mass (monthly mean). (Left column) reference simulation (Aging case), (right column) difference between reference and No Aging case. Note the inverted color scales for the bottom two panels, where higher aerosol water mass is shown in blue and lower in red.

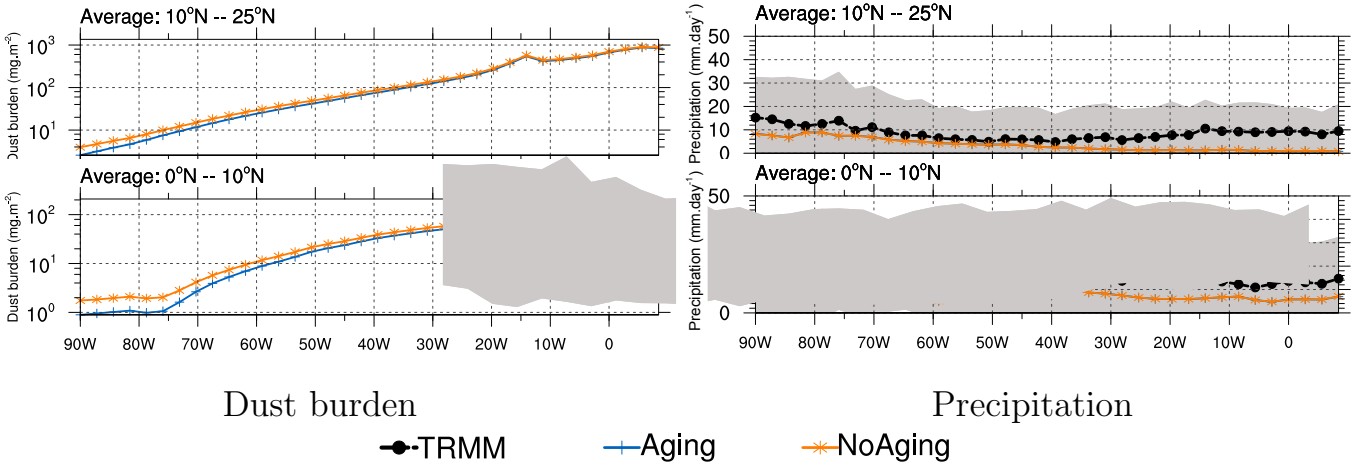

**Figure 15.** (Left) Dust burden, (right) precipitation for different regions: (Top) dust transport over the Atlantic Ocean zone, (bottom) dust-ITCZ zone $0°$to $10°$N. The shaded area represents one standard deviation of TRMM precipitation. The results show the long-term average of the entire evaluation period 2000-2012.