# Peer review of "Sensitivity of transatlantic dust transport to chemical aging and related atmospheric processes"

_Atmospheric Chemistry and Physics, 2016_

## Referee Comment (RC1) · Anonymous Referee #1 · 25 Aug 2016

Chemical aging of atmospheric mineral dust during transatlantic transport Abdelkader et al.,

The authors used EMAC (The ECHAM5/MESSy2 atmospheric chemistry General Circulation Model) to evaluate transport and loadings of mineral dust particles during transatlantic transport. The study carefully considered aging mineral dust in the model and compared the results with non-aging mineral dust particles. They found some interesting results such as the removable efficiency and optical properties. These results will be potential useful for the future study on the ground base. On the other hand, the study carefully used the satellite data (AOD and CALIPSO) to calibrate the modeling results. They obtained the consistent results. The developed method is significant to improve the current model.

The mineral dust particles are important for climate change, biogeochemical cycle, and heterogeneous atmospheric chemistry in global. Many studies found how the mineral dust changes in air. However, the modeling work is rare. The modeling work is useful to evaluate effects of mineral dust in the air. Although the modeling parameters are not based on measurements, the results and comparison is interesting.

I would like to recommend accepting this paper after one minor revision. In the introduction section, the authors should add some findings in field campaigns which have revealed the nitrate coatings on alkaline mineral dust particles in the worlds. For example, Tobo, Li, Sullivan et al., found mineral dust aging process in the air. Although the authors consider the mineral dust particles absorbing acidic gases transformed from $SO_2$, $NO_x$, or HCl. However, these field study all pointed out the nitrate coating determine particle hygroscopic properties ("Asian dust particles converted into aqueous droplets under remote marine atmospheric conditions." P Natl Acad Sci USA 107(42): 17905-17910./ "Observation of nitrate coatings on atmospheric mineral dust particles." Atmos. Chem. Phys. 9(6): 1863-1871/"Direct observations of the atmospheric processing of Asian mineral dust." Atmos. Chem. Phys. 7: 1213-1236.).

The authors should mention the aged mineral dust particles become hydrophilic and can act as CCN during the transport (Mixing state and hygroscopicity of dust and haze particles before leaving Asian continent. J. Geophys. Res. 119 (2), 1044-1059.). Page 1 line 16 miss blank after comma.

Figure 3 should be marked where is the Cribbean

Section 4.3 Why did not the authors consider the mineral dust as ice nucleation? It could be one removable pathway for mineral dust in air.

I recommend revising the current title. Because the study focused on the evaluation of mineral dust during transatlantic transport using model and other methods, it didn't study chemical aging of mineral dust. The current title seems that the study understand the chemical aging mechanism of mineral dust in the air.

---

## Referee Comment (RC2) · Anonymous Referee #2 · 9 Sep 2016

General comments

This work describes the effects of chemical aging, emissions and convection parameterizations in the transport of desert dust over the Atlantic Ocean with the use of the atmospheric chemistry general circulation model EMAC. The authors have published the concept of dust chemical aging in a recent paper and in this new publication they deal with the transatlantic transport and how it can be affected by various model parameterizations related to the dust cycle. Modeling the desert dust cycle is a complicated topic given the necessity to parameterize physical processes that produce and cycle dust particles throughout the atmosphere and a better understanding of how to improve these processes is significant.

I found the paper difficult to read, in terms of the flow, especially because there is a

continuous description of the figures instead of using them to support a conclusion or remark. The main review comments are related to clarifications in the methodology and discussion of the results. I am in favor of publishing this paper with Atmospheric Chemistry and Physics with Major Revisions. The specific comments that follow will help improve the discussion of the methodology and significance of the findings so that the overall quality of the manuscript is enhanced.

Specific comments/suggestions

1. Please refer to aging of dust as "chemical aging" in all parts of the manuscript.

2. Introduction, page 2, line 34: in the sentence "mean normalized bias of the AOD model varies", the word "model" should be omitted.

3. Please provide the specific modules used in the EMAC configuration so that the results from this work can be reproducible.

4. Are indirect aerosol-cloud interactions included in the model configuration, besides the radiative feedback effect? How different the results might have been if these interactions were included?

5. Page 3, line 23: what is the meaning of "increases the level of dust aging"? Is there a specification of levels of chemical aging that the authors consider? I am assuming that inorganic acids uptake by the dust particles is what differentiates freshly emitted dust with dust being transported in the atmosphere, which eventually leads to "chemical aging" since the original dust particle has an altered chemical signature. Unless water uptake is considered the primary aging process. Please clarify.

6. Following the same notion as in comment #4, Figure 1 indicates that insoluble emitted dust turns into aged-dust, followed by acid condensation. I would expect the acid condensation first and then the dust characterized as aged. Based on this schematic, there is no clear distinction about when dust is termed aged or non-aged.

7. Page 3, line 27: "the mineral cations are used as reactivity proxy for natural aerosols,

such as [. . .] mineral dust". Knowing how difficult it is to include chemical speciation of the emitted dust particles in the model, my question is how the authors apportioned the dust emitted mass to mineral cations. Is it a fixed percentage for calcium, magnesium and potassium? This information must be made clear in the text.

8. Sections 3 and 4: as mentioned in the beginning, in a lot of parts of the discussion there is a description of the figure instead of a narrative about the main findings, followed and supported by the figures. I strongly encourage the authors to revise parts of the text accordingly, which will greatly benefit the quality of the manuscript.

9. What is the basis for the selection of the six specific stations that were included in the sensitivity tests, out of the ones shown in Fig. 4? It seems from fig.4 that more stations were available inside the specific zones.

10. Page 6, line 9: is the 600ug/m3 an observed or simulated value for dust concentration?

11. Page 7, lines 2-3: the aging of dust particles throughout the transatlantic transport depends also on the availability of inorganic acids in this region. The EMAC model outputs corroborate with the assumption that inorganic acids can be found in the DTA and/or DIZ zone?

12. Table 1: I believe rm and ro are supposed to be standard deviations sigmam and sigmao. Please revise accordingly.

13. Table 1: what is GFE, PF2 and PF10? They are not included in the appendix and never mentioned in the text.

14. Figure 10, caption: please include the time period that the plots cover. Also, remind the reader which plots correspond to the ECMWF and TIEDTKE schemes.

15. Figure 11: is the standard deviation of the TRMM product calculated over the meridional mean to show the variation/dispersion of the precipitation at each longitude? Why not show the stdev for the model outputs as well?

16. Are Figures 10 and 11 for the same time period, July 2009? If so, the meridional means are confusing. They show that B1T5 is closer to the observations but Figure 10 indicates that maybe EMAC base case is closer to TRMM.

17. Figures 13 and 14 show monthly means for July 2009?

18. The paper title in the supplement is not correct. Please revise accordingly.

19. In the conclusions section, there is discussion on the findings from the sensitivity tests and model evaluation. A general conclusion about the new and significant findings from this work is necessary and, perhaps, a recommendation to the model users about the choices that would produce more reliable mineral dust simulations.

---

## Referee Comment (RC3) · Anonymous Referee #3 · 14 Sep 2016

General comments:

The abstract and the body of the text are not consistent, and the text does not efficiently support the conclusions in the abstract. In fact, the abstract and the text look like parts of different papers.

There are two major results in the abstract. One result is on the pattern of dust transport over the Atlantic, which is characterized by (1) a steep and linear westward gradient due to the dust sedimentation (dry deposition) in the DTA zone and (2) an efficient removal dominated by cloud interaction and wet deposition in the DIZ zone. Another result is on the aging process of dust particles and on the effect of the aging on dust AOD in addition to the removal of the dust. About the later result, authors give the details as (1) aging of dust particles by absorbing inorganic acids changes the particles

into soluble modes, enhances the absorption of water vapor, and consequently causes the increase of AOD, which the authors name as "direct effect of dust aging", and (2) aging of dust particles causes more efficient removal of particles in comparison with non-aged dust particles, and consequently results in a decrease of dust AOD, which the authors name as "indirect effect of dust aging".

However, the text of results and discussion in the manuscript does not focus on the above two results. Here are my understandings on the text.

Section 3: In the first part (Figure 2, and also Figure 3, which is somewhat a repeat of Figure 2), the simulated result (first result mentioned above) and the possible reasons for the result are simply introduced and described. As a major result of this study described in the abstract, more details and a deep discussion are necessary. My major concern on this part is the lack of a discussion on the uncertainties in the result. Another concern is that this part is not consistent with the purpose of this Section, which is to evaluate the performance of the model (the first line of Section 3). The remaining parts of this section are the evaluation of the model performance with the comparison to AERONET observations.

Section 4: This part is an evaluation of model performance, too. First, the evaluation is conducted with a focus on the model sensitivity to emission flux and to removal mechanisms. Then the influence of different convection schemes and dust chemical aging on simulation results is examined.

Although the major results described in the abstract are introduced in Section 3 and Section 4, the results are not described in a clear and compacted way. In addition, the explanations of the consistence and difference between the simulation results and the observational facts are very qualitative and the uncertainties are not quantitatively discussed.

The evaluation of the model performance is not bad and is acceptable. But the evaluation shows the quality of the model and has a weak relation with the conclusions

described in the abstract.

So the contents of abstract are inconsistent with the contents of results and discussion (Section 3 and Section 4). Actually, many parts in the text of results and discussions are repeats of the paper of Abdelkader et al. (2015). The first result described in the abstract is original in this model study, but the second result contains less new information in comparison with Abdelkader et al. (2015).

Other major comments:

. The abstract is tedious and hardly followed.

. Figure 1 is not necessary according to the abstract. The model has been described and evaluated in Abdelkader et al. (2015).

. Removal processes of dust particles by dry and wet deposition, including the subsequences of dust aging, are repeatedly applied to explain simulation results. In addition to that the repeats make the manuscript very tedious, almost all explanations lack of a discussion on the confidence of the explanations, i.e. to what a degree the explanations can account for the results. Discussions with quantitative evaluation are necessary to increase the quality of the explanations.

. The description on the wet deposition of dust particles associated with the aging of particles lacks of details and is not clear. The removal is simply described as the processes of the hygroscopic growth of aged particles (Section 4.3) and is discussed with comparisons associated with precipitation (convection) and dust emission (Section 4.2). Hygroscopic growth is a subsequence of particle aging (i.e. interaction with cloud), which is emphasized in this manuscript. However, precipitation is fundamentally governed by thermodynamic properties and the movement of air parcels (the convection: Aerosol particles are not included in the simulation of water vapor distributions by Tost et al. (2006b)). Precipitation removes dust particles via the adoption of dust particles by cloud droplets and raindrops in cloud and in below-cloud air (the effect of

washout) and/or via the raincloud droplet formation on dust particles under saturate conditions in cloud or the adjacent air (the effect of nucleation scavenging). The two scavenging processes are closely dependent on the size of particles and droplets. Under saturate conditions (in cloud), dust-induced droplets (nucleation scavenging) may grow into a large droplets. But the size, rather than the composition, of a particle is the key factor for the nucleation at the size range of dust particles, usually larger than several hundred nano-meters (Dusek et al. 10.1126/science.1125261, Science, Vol. 312, Issue 5778, pp. 1375-1378). In below-cloud air under sub-saturate conditions, the growth of aged particles due to water vapor absorption is limited and the particles are not expected to frequently become considerable larger than the original particles. So the relative importance of the two processes in the dust removal needs to be clearly described and discussed in order to quantitatively show how important of the subsequence of dust aging is and how the aging enhances the removal of aged dust particles. It sounds that washout is not important for the removal of the dust particles in DIZ zone. Is this correct?

. The definition of "direct effect" and "indirect effect" of dust aging needs to be carefully re-considered. In this study, the effect is limited to that on AOD. However, there are many other effects associated with the aging, such as the absorption of acid gaseous species and the change of gas phase reactions. In addition, the definition may cause a confusion when readers think the "direct and indirect climate effects of aerosol particles".

---

## Author Response (AR1)

**Chemical aging of atmospheric mineral dust during transatlantic transport**

**Reply to Anonymous Referee #1 (doi:10.5194/acp-2016-470-RC1)**

http://editor.copernicus.org/index.php/acp-2016-470-RC1.pdf?\_mdl=msover\_md&\_jrl=10&\_lcm=oc108lcm109w&\_acm=get\_comm\_file&\_ms=51800&c=111158&salt=895046841231028569

by Mohamed Abdelkader and Swen Metzger, et al.,

December 19, 2016

We thank the anonymous referee for the comments on this manuscript. The comments and questions raised are addressed below by our point-by-point reply (black) and the revised MS.

The authors used EMAC (The ECHAM5/MESSy2 atmospheric chemistry General Circulation Model) to evaluate transport and loadings of mineral dust particles during transatlantic transport. The study carefully considered aging mineral dust in the model and compared the results with nonaging mineral dust particles. They found some interesting results such as the removable efficiency and optical properties. These results will be potential useful for the future study on the ground base. On the other hand, the study carefully used the satellite data (AOD and CALIPSO) to calibrate the modeling results. They obtained the consistent results. The developed method is significant to improve the current model.

We thank the referee for this general comment.

The mineral dust particles are important for climate change, biogeochemical cycle, and heterogeneous atmospheric chemistry in global. Many studies found how the mineral dust changes in air. However, the modeling work is rare. The modeling work is useful to evaluate effects of mineral dust in the air. Although the modeling parameters are not based on measurements, the results and comparison is interesting.

We also appreciate this comment.

I would like to recommend accepting this paper after one minor revision. In the introduction section, the authors should add some findings in field campaigns which have revealed the nitrate coatings on alkaline mineral dust particles in the worlds. For example, Tobo, Li, Sullivan et al., found mineral dust aging process in the air. Although the authors consider the mineral dust particles absorbing acidic gases transformed from SO2, NOx, or HCl. However, these field study all pointed out the nitrate coating determine particle hygroscopic properties ("Asian dust particles converted into aqueous droplets under remote marine atmospheric conditions." P Natl Acad Sci USA 107(42): 17905-17910./ "Observation of nitrate coatings on atmospheric mineral dust particles." Atmos. Chem. Phys. 9(6): 1863-1871/"Direct observations of the atmospheric processing of Asian mineral dust." Atmos. Chem. Phys. 7: 1213-1236.).

We do agree that nitrate coating can determine hygroscopicity of mineral dust particles, which

is especially the case in an polluted atmosphere (Bauer et al., 2007). Moreover, our EMAC setup accounts for this effect, since the nitric acid (e.g., as oxidation end product of combustion  $NO_x$ ) may react in our set-up with the calcium fraction of the mineral dust particles to form calcium nitrate, which takes up water vapour from the atmosphere at ambient conditions where the humidity is just about 50% (the RHD of Ca(NO3)2 is 48% at T=298 K). In strong contrast, dust coating by sulphuric acid does not lead to hygroscopic particles since the RHD of CaSO4 is close to 100% (at any T).

The authors should mention the aged mineral dust particles become hydrophilic and can act as CCN during the transport (Mixing state and hygroscopicity of dust and haze particles before leaving Asian continent. J. Geophys. Res. 119 (2), 1044-1059.)

This sentence and the reference has been added to the introduction of the revised MS.

Page 1 line 16 miss blank after comma.

We have added this blank in the revised MS.

Figure 3 should be marked where is the Cribbean.

We have marked the Caribbean in Figure 3 of the revised MS.

Why did not the authors consider the mineral dust as ice nucleation? It could be one removable pathway for mineral dust in air.

We agree that the consideration of mineral dust can be regionally important for ice nucleation. However, the effect will be less pronounced for our global modeling. The main reason is simply that the cloud micro-physical processes needs to be parameterized for the still relatively coarse model grid box (here approx 110 km). On these (model grid) scales many (partly unknown) micro-physical processes are implicitly parameterized, if the model results more or less agree with e.g., AOD observations. Changes in the micro-physical assumptions will therefore not alter the overall picture much. We have learned that from several additional sensitivity studies. Thus, for the current scope of this paper, we omit a more explicit aerosol-cloud coupling that includes feedback of mineral dust particles on ice nucleation. Aerosol-cloud coupling of dust is implicitly accounted for by changes in solubility of the aged dust particles due water uptake, which feeds back with scavenging, cloud water content, remaining aerosol loadings and radiation. A more detailed analysis of the current assumptions on aerosol-cloud coupling will be presented elsewhere.

I recommend revising the current title. Because the study focused on the evaluation of mineral dust during transatlantic transport using model and other methods, it didn?t study chemical aging of mineral dust. The current title seems that the study understand the chemical aging mechanism of mineral dust in the air.

We have revised the title to: "Sensitivity of transatlantic dust transport to chemical aging and related atmospheric processes".

**References**

Bauer, S. E., Mishchenko, M. I., Lacis, A. A., Zhang, S., Perlwitz, J., and Metzger, S. M.: Do sulfate and nitrate coatings on mineral dust have important effects on radiative properties and climate modeling?, Journal of Geophysical Research: Atmospheres, 112, D06307, doi: 10.1029/2005JD006977, URL http://dx.doi.org/10.1029/2005JD006977, 2007.

**Chemical aging of atmospheric mineral dust during transatlantic transport**

**Reply to Anonymous Referee #2 (doi:10.5194/acp-2016-470-RC2)**

http://editor.copernicus.org/index.php/acp-2016-470-RC2.pdf?\_mdl=msover\_md&\_jrl=10&\_lcm=oc108lcm109w&\_acm=get\_comm\_file&\_ms=51800&c=111819&salt=102505675193170096

by Mohamed Abdelkader and Swen Metzger, et al.,

December 19, 2016

We thank the anonymous referee for comments on this manuscript. The comments and questions raised are addressed below by our point-by-point reply (black) and the revised MS.

General comments

This work describes the effects of chemical aging, emissions and convection parameterizations in the transport of desert dust over the Atlantic Ocean with the use of the atmospheric chemistry general circulation model EMAC. The authors have published the concept of dust chemical aging in a recent paper and in this new publication they deal with the transatlantic transport and how it can be affected by various model parameterizations related to the dust cycle. Modeling the desert dust cycle is a complicated topic given the necessity to parameterize physical processes that produce and cycle dust particles throughout the atmosphere and a better understanding of how to improve these processes is significant.

We thank the referee for this general comment.

I found the paper difficult to read, in terms of the flow, especially because there is a continuous description of the figures instead of using them to support a conclusion or remark. The main review comments are related to clarifications in the methodology and discussion of the results. I am in favor of publishing this paper with Atmospheric Chemistry and Physics with Major Revisions. The specific comments that follow will help improve the discussion of the methodology and significance of the findings so that the overall quality of the manuscript is enhanced.

We also appreciate the specific comments.

Specific comments/suggestions.

1. Please refer to aging of dust as "chemical aging" in all parts of the manuscript.

Changed "aging" to "chemical aging" throughout the manuscript.

2. Introduction, page 2, line 34: in the sentence "mean normalized bias of the AOD model varies", the word "model" should be omitted.

The word "model" is omitted.

3. Please provide the specific modules used in the EMAC configuration so that the results from this work can be reproducible.

Table 1 is included which shows the EMAC submodels used in the study.

**4. Are indirect aerosol-cloud interactions included in the model configuration, besides the radiative feedback effect? How different the results might have been if these interactions were included?**

Yes, through changes in the scavenging efficiency, but not through changes in CCN activity. The impact of the latter does not alter our results, since we have focused on the chemical aging of a major dust outflow between 2000 and 2013 (i.e., July 2009). For such a case, the chemical aging as represented here (various effects of changes in the wet radius) dominates the aerosol-cloud-radiation coupling. Nevertheless, the topic deserves further investigations and will be subject of a follow-up study, which then will focus on the chemical aging of weaker dust-outflow events.

5. Page 3, line 23: what is the meaning of "increases the level of dust aging"? Is there a specification of levels of chemical aging that the authors consider? I am assuming that inorganic acids uptake by the dust particles is what differentiates freshly emitted dust with dust being transported in the atmosphere, which eventually leads to "chemical aging" since the original dust particle has an altered chemical signature. Unless water uptake is considered the primary aging process. Please clarify.

We have changed this sentence to: "This increases the dust particle mass, particle size and the removal rates, which tends to decrease the lifetime of chemically aged dust."

6. Following the same notion as in comment #4, Figure 1 indicates that insoluble emitted dust turns into aged-dust, followed by acid condensation. I would expect the acid condensation first and then the dust characterized as aged. Based on this schematic, there is no clear distinction about when dust is termed aged or non-aged.

Indeed. Figure 1 has been revised accordingly.

7. Page 3, line 27: "the mineral cations are used as reactivity proxy for natural aerosols, such as [. . .] mineral dust". Knowing how difficult it is to include chemical speciation of the emitted dust particles in the model, my question is how the authors apportioned the dust emitted mass to mineral cations. Is it a fixed percentage for calcium, magnesium and potassium? This information must be made clear in the text.

Yes, we follow Abdelkader et al. (2015) and use a fixed percentage for this study. This percentage has been determined in order to best match the observations of various mineral cations from EMEP and CASTNET observations. A more comprehensive treatment is under development.

8. Sections 3 and 4: as mentioned in the beginning, in a lot of parts of the discussion there is a description of the figure instead of a narrative about the main findings, followed and supported by the figures. I strongly encourage the authors to revise parts of the text accordingly, which will greatly benefit the quality of the manuscript. Both parts have been revised.

9. What is the basis for the selection of the six specific stations that were included in the sensitivity tests, out of the ones shown in Fig. 4? It seems from fig.4 that more stations were available inside the specific zones.

Figure 4 includes the stations that have data for a longterm evaluation (2000-2012), while only the selected stations have observations for the selected period (July 2009).

10. Page 6, line 9: is the 600ug/m3 an observed or simulated value for dust concentration?. The values refers to the model. We have added "modeled surface concentration" for clarification.

11. Page 7, lines 2-3: the aging of dust particles throughout the transatlantic transport depends also on the availability of inorganic acids in this region. The EMAC model outputs corroborate with the assumption that inorganic acids can be found in the DTA and/or DIZ zone?

Yes. The inorganic aerosol precursor gases (HCl,  $HNO_3$ ,  $H_2SO_4$ ) are ubiquitous, as we consider in our EMAC study various processes and anthropogenic (e.g., ships and flight traffic) and natural sources (e.g., lighting, chlorine activation of sea spray due condensation of e.g.,  $HNO_3$ ,  $H_2SO_4$ ).

12. Table 1: I believe rm and ro are supposed to be standard deviations sigmam and sigmao. Please revise accordingly.

 $\mathbf{r}_m$  and  $\mathbf{r}_o$  are the geometric mean of the model and observations, respectively. We have added a description of the statistical parameters in the Appendix A.

13. Table 1: what is GFE, PF2 and PF10? They are not included in the appendix and never mentioned in the text.

GFE denotes the Growth Factorial Error, while PF2 the Fractions of points within a factor of two from the observations; accordingly, PF10, the points within a fraction of 10 from the observations. The definitions have been added to the description in Appendix A.

14. Figure 10, caption: please include the time period that the plots cover. Also, remind the reader which plots correspond to the ECMWF and TIEDTKE schemes.

The time period is now included in the figure captions.

15. Figure 11: is the standard deviation of the TRMM product calculated over the meridional mean to show the variation/dispersion of the precipitation at each longitude? Why not show the stdev for the model outputs as well?

Yes. The standard deviation of the model results has been included in Figure 11.

16. Are Figures 10 and 11 for the same time period, July 2009? If so, the meridional means are confusing. They show that B1T5 is closer to the observations but Figure 10 indicates that maybe EMAC base case is closer to TRMM.

Yes, both figures show monthly averages for July 2009. But, comparing Figure 10 and 11 is somewhat deceptive, since Figure 10 represents a qualitative comparison of the spatial distribution of precipitation and the extent of the dust plume, while Figure 11 represents a quantitate comparison, which generally is more accurate. And from Figure 11 the simulation B1T5 is closer to the observations, at least for 90-50W, while the opposite is only true for the region of 20-10W.

17. Figures 13 and 14 show monthly means for July 2009?

Yes, this is now noted in the figure caption.

18. The paper title in the supplement is not correct. Please revise accordingly.

Both changed, according to the comment of reviewer one to: "Sensitivity of transatlantic dust transport to chemical aging and related atmospheric processes".

19. In the conclusions section, there is discussion on the findings from the sensitivity tests and model evaluation. A general conclusion about the new and significant findings from this work is necessary and, perhaps, a recommendation to the model users about the choices that would produce more reliable mineral dust simulations.

A general conclusion and a recommendation has been added to the conclusions section.

December 19, 2016

We thank the anonymous referee for the in-depth comments on this manuscript. The comments and questions raised are addressed below by our point-by-point reply (black) and the revised MS.

**General comments**

The abstract and the body of the text are not consistent, and the text does not efficiently support the conclusions in the abstract. In fact, the abstract and the text look like parts of different papers.

There are two major results in the abstract. One result is on the pattern of dust transport over the Atlantic, which is characterized by (1) a steep and linear westward gradient due to the dust sedimentation (dry deposition) in the DTA zone and (2) an efficient removal dominated by cloud interaction and wet deposition in the DIZ zone. Another result is on the aging process of dust particles and on the effect of the aging on dust AOD in addition to the removal of the dust. About the later result, authors give the details as (1) aging of dust particles by absorbing inorganic acids changes the particles into soluble modes, enhances the absorption of water vapor, and consequently causes the increase of AOD, which the authors name as "direct effect of dust aging", and (2) aging of dust particles causes more efficient removal of particles in comparison with non-aged dust particles, and consequently results in a decrease of dust AOD, which the authors name as "indirect effect of dust aging". However, the text of results and discussion in the manuscript does not focus on the above two results.

The abstract and the discussion in the manuscript have been revised accordingly.

Here are my understandings on the text. Section 3: In the first part (Figure 2, and also Figure 3, which is somewhat a repeat of Figure 2), the simulated result (first result mentioned above) and the possible reasons for the result are simply introduced and described. As a major result of this study described in the abstract, more details and a deep discussion are necessary. My major concern on this part is the lack of a discussion on the uncertainties in the result. Another concern is that this part is not consistent with the purpose of this Section, which is to evaluate the performance of the model (the first line of Section 3). The remaining parts of this section are the evaluation of the model performance with the comparison to AERONET observations.

The text has been revised to be consistent with the purpose of this section and a note on the

uncertainties of the result has been added. For a discussion on the uncertainties we refer, however, to Section 4 "Sensitivity studies", since this section is exactly about the modeling uncertainties.

Section 4: This part is an evaluation of model performance, too. First, the evaluation is conducted with a focus on the model sensitivity to emission flux and to removal mechanisms. Then the influence of different convection schemes and dust chemical aging on simulation results is examined. Although the major results described in the abstract are introduced in Section 3 and Section 4, the results are not described in a clear and compacted way. In addition, the explanations of the consistence and difference between the simulation results and the observational facts are very qualitative and the uncertainties are not quantitatively discussed.

The text has been revised such that this section 4 "Sensitivity studies", now clearly deals with modeling uncertainties (and not again of model performance evaluation).

The evaluation of the model performance is not bad and is acceptable. But the evaluation shows the quality of the model and has a weak relation with the conclusions described in the abstract.

The conclusions and the abstract have been revised accordingly.

So the contents of abstract are inconsistent with the contents of results and discussion (Section 3 and Section 4). Actually, many parts in the text of results and discussions are repeats of the paper of Abdelkader et al. (2015). The first result described in the abstract is original in this model study, but the second result contains less new information in comparison with Abdelkader et al. (2015).

The study of Abdelkader et al. (2015) presents the dust-air pollution interaction over the Easter Mediterranean, while this work focuses on a "Sensitivity of transatlantic dust transport to chemical aging and related atmospheric processes" – the new title (see our reply to referee #1). Since both studies focus on the chemical aging of dust, there is of course some overlap in the description. Otherwise this paper would not be able to stand alone. To our opinion, the overlap is small and important to have for the average reader to understand the main text flow without referring to Abdelkader et al. (2015), which an interested reader of course will/shall do.

Other major comments The abstract is tedious and hardly followed.

The abstract has been revised.

Figure 1 is not necessary according to the abstract. The model has been described and evaluated in Abdelkader et al. (2015).

We prefer to have this paper a standalone (see our above) and, hence, we keep Figure 1.

Removal processes of dust particles by dry and wet deposition, including the subsequences of dust aging, are repeatedly applied to explain simulation results. In addition to that the repeats make the manuscript very tedious, almost all explanations lack of a discussion on the confidence of the explanations, i.e. to what a degree the explanations can account for the results. Discussions with quantitative evaluation are necessary to increase the quality of the explanations.

Redundancies have been removed and an extended discussion on a more quantitative evaluation has been included based on the statistical parameters shown in Table 1a,b of the Supplement.

The description on the wet deposition of dust particles associated with the aging of particles lacks of details and is not clear. The removal is simply described as the processes of the hygroscopic growth of aged particles (Section 4.3) and is discussed with comparisons associated with precipitation (convection) and dust emission (Section 4.2). Hygroscopic growth is a subsequence of particle aging (i.e. interaction with cloud), which is emphasized in this manuscript. However, precipitation is fundamentally governed by thermodynamic properties and the movement of air parcels (the convection: Aerosol particles are not included in the simulation of water vapor distributions by Tost et al. (2006b)). Precipitation removes dust particles via the adoption of dust particles by cloud droplets and raindrops in cloud and in below-cloud air (the effect of washout) and/or via the raincloud droplet formation on dust particles under saturate conditions in cloud or the adjacent air (the effect of nucleation scavenging). The two scavenging processes are closely dependent on the size of particles and droplets. Under saturate conditions (in cloud), dust-induced droplets (nucleation scavenging) may grow into a large droplets. But the size, rather than the composition, of a particle is the key factor for the nucleation at the size range of dust particles, usually larger than several hundred nano-meters (Dusek et al. 10.1126/science.1125261, Science, Vol. 312, Issue 5778, pp. 1375-1378). In below-cloud air under sub-saturate conditions, the growth of aged particles due to water vapor absorption is limited and the particles are not expected to frequently become considerable larger than the original particles. So the relative importance of the two processes in the dust removal needs to be clearly described and discussed in order to quantitatively show how important of the subsequence of dust aging is and how the aging enhances the removal of aged dust particles. It sounds that washout is not important for the removal of the dust particles in DIZ zone. Is this correct?.

No, the washout is of course also important for the removal of the dust particles in DIZ zone, but the chemical aging and scavenging of aged dust particles are according to our study more important in the DIZ–zone compared to DTA–zone. The text has been revised accordingly.

The definition of "direct effect" and "indirect effect" of dust aging needs to be carefully reconsidered. In this study, the effect is limited to that on AOD. However, there are many other effects associated with the aging, such as the absorption of acid gaseous species and the change of gas phase reactions. In addition, the definition may cause a confusion when readers think the "direct and indirect climate effects of aerosol particles".

We do agree that the "direct effect" and "indirect effect" of chemical aging of dust seems limited only by a definition of AOD, but it actually includes all other effects. Indeed, we try to limit the definition to the AOD, since only the net-effect AOD eventually drives the radiation. Of course, the total effect includes many other processes, such as heterogeneous reactions on dust particles, which can either increase of decrease the AOD. But, at the end of a computation step only the net-effect on AOD accounts. Therefore, we keep our definitions as introduced here.

[revised manuscript text omitted]
 stronger during July 2009 ( $\approx 2 \text{ g m}^{-2}$  compared to  $\approx 0.2 \text{ g m}^{-2}$  during July 2009), whereas the respectively). The elevated precipitation over the Caribbean shows the maximum dust depositiondue to scavengingcauses maximum wet deposition. As a result, the dust burden is an order of magnitude lower over the Caribbean compared with to West Africa. In addition, there is a clear anticorrelation anti-correlation between the dust burden and the precipitation amount over both sides
- 30 of the Atlantic. The comparison of precipitation with TRMM observations shows reveals that the EMAC model gives more realistic results over West Africa compared with the Caribbean for all convection schemes.

The Second, the ADP (Fig. 11) illustrates the effect of convection schemes on the transatlantic dust transportand shows the highest sensitivity for the convection parameterization. Over West Africa, the dust is already aged with ADP values between 0.2 and 0.4, whereas over the Caribbean the ADP values are higher ranging between with 0.3 and 0.5 only slightly higher. The

35 lower ADP values over West Africa indicate can be attributed to the higher dust loadings, which require a requires a much

larger amount of condensable material to agebecomes fully aged. Over the Caribbean, the dust loading is much considerably lower due to removal during the transport which is the removal processes along dust transport, which takes about 5 days, for instance Gläser et al. (2015), long enough. This time is sufficiently long for coating by acids and other soluble materials which cause (Gläser et al., 2015), and causes the dust to become more aged(ADP = 0.6) compared to the Western African

- 5 side (ADP=0.35). The . On the other hand, the high precipitation amount at 15°W over the Western Africa region results in higher scavenging of the aged dust particles compared with the "pristine" (nonaged-pristine- (non-aged) dust particlesand. This results in a decrease in the ADP valuesthat are , in agreement with the results of Abdelkader et al. (2015). Western to West of 15°W, the dust is transported over the Atlantic at into a region where the precipitation is precipitation is much lower (middle panels). This results in an increase in the aging levels. Consequently, the level of chemical aging increases. The EMAC
- 10 reference simulation (with higher precipitation) shows too strong precipitation) therefore shows a higher ADP (0.35 compared to 0.2) values as, which is a result of the lower dust burden, which is caused by and mainly caused by a too efficient wet removal.

The Thus, the convection sensitivity analysis indicates a very strong removal of the dust during transatlantic transport with the EMAC points to a too strong removal mechanism of the mineral dust particles along transatlantic transport, when the default

- 15 convection scheme , which is indicated by the underestimation of the AOD over the Caribbeanis used in EMAC. In addition, the level of dust aging controls chemical aging seems to control the efficiency of dust scavenging. Higher levels of aged dust, and higher precipitation amounts, significantly decrease the dust burden and thus the AOD over the Caribbean. This suggests that improving further suggests that modeling the transatlantic dust transports requires improved convection parameterization and (i.e., more realistic precipitation rates in parallel with the improved dust), and in parallel a realistic representation of dust
- 20 chemical aging.

**4.3 Dust chemical aging**

The level of dust aging depends on the availability of inorganic acids, i.e., volatile and semivolatile compounds. To further investigate the impact of the dust chemical aging on the transatlantic dust transport, dust aging this process was excluded for an additional sensitivity study. For this The level of dust chemical aging depends on the availability of condensable acids (see

- 25 Sec. 2). For the "No Aging" case, the condensation of acids on insoluble dust particles is excluded, which suppresses water uptake by dust particles. Figure 12 shows the AOD time series at the AERONET stations on both sides of the Atlantic for the two cases, i.e., "Aging " and "No Aging" Aging and No Aging. Generally, the "Aging " Aging case systematically shows a higher AOD as compared with the "No Aging " compared to the No Aging case, which emphasizes the importance of this process and the associated water uptake in agreement with the results of Pozzer et al. (2015). The dust Abdelkader et al. (2015).
- 30 However, the dust chemical aging has a stronger impact on the AOD over Western West Africa, especially at the Capo Verde and Dakar stations during the two dust outbreaks discussed above. The Aging case shows about 0.2 higher AOD compared with the "No Aging " No Aging case as a result of the larger particle size and the associated water uptake. This increases the scattering cross section and thus the AOD. Over the Caribbean, the dust chemical aging shows a smaller impact on the AOD; the "Aging " Aging case shows only about 0.05 higher AOD because of the lower contribution of the dust to the overall AOD

values (which includes the contribution of other aerosol species, sea salt, etc., for instance). During the high dust outbreaks, the concentration of the soluble compounds required to coat such a large amount of dust is not available according to the EMAC model. The aged dust particles are removed more efficiently during transport and relatively more uncoated dust particles reach the Caribbean. As a result, the dust chemical aging has a limited effect on the AOD over the Caribbean AERONET stations.

- 5 Figure 13 shows the regional difference (monthly mean) for (a) the dust burden, (b) AOD, (c) dust emissions averaged over the region from 18°-22°N, and (d) the dust-only AOD ("No Aging " minus "Aging " No Aging minus Aging case). The results show a higher dust burden over the dust-source regions in Western-West Africa for the "No Aging " case as No Aging case compared with the reference case ("Aging" case(Aging). For the "No Aging " 
[revised manuscript text omitted]

As EMAC, factor=2.61 in the horizontal flux
Reference simulation; TIEDTKE convection with NORDENG closure                                                                                                                                                                                                                                                                                                                     |
|            | B1E7
B1E8
EMAC
B1T2                                          | As EMAC, the accumulation and the coarse modes increased by a factor of 2.61
As EMAC, factor=2.61 in the horizontal flux
Reference simulation; TIEDTKE convection with NORDENG closure
TIEDTKE convection with TIEDTKE closure (Tiedtke, 1989)                                                                                                                                                                                                                                                          |
| Convection | B1E7
B1E8
EMAC
B1T2
B1T3                                  | As EMAC, the accumulation and the coarse modes increased by a factor of 2.61
As EMAC, factor=2.61 in the horizontal flux
Reference simulation; TIEDTKE convection with NORDENG closure
TIEDTKE convection with TIEDTKE closure (Tiedtke, 1989)
TIEDTKE convection with HYBRID closure (Tiedtke, 1989)                                                                                                                                                                                                |
| Convection | B1E7         B1E8         EMAC         B1T2         B1T3         B1T4 | As EMAC, the accumulation and the coarse modes increased by a factor of 2.61
As EMAC, factor=2.61 in the horizontal flux
Reference simulation; TIEDTKE convection with NORDENG closure
TIEDTKE convection with TIEDTKE closure (Tiedtke, 1989)
TIEDTKE convection with HYBRID closure (Tiedtke, 1989)
ECMWF operational convection scheme (Bechtold et al., 2004)                                                                                                                                 |
| Convection | B1E7         B1E8         EMAC         B1T2         B1T3         B1T4 | As EMAC, the accumulation and the coarse modes increased by a factor of 2.61
As EMAC, factor=2.61 in the horizontal flux
Reference simulation; TIEDTKE convection with NORDENG closure
TIEDTKE convection with TIEDTKE closure (Tiedtke, 1989)
TIEDTKE convection with HYBRID closure (Tiedtke, 1989)
ECMWF operational convection scheme (Bechtold et al., 2004)
with the shallow convection closure of Grant and Brown (1999)                                                                |
| Convection | B1E7
B1E8
EMAC
B1T2
B1T3
B1T4
B1T5                  | As EMAC, the accumulation and the coarse modes increased by a factor of 2.61
As EMAC, factor=2.61 in the horizontal flux
Reference simulation; TIEDTKE convection with NORDENG closure
TIEDTKE convection with TIEDTKE closure (Tiedtke, 1989)
TIEDTKE convection with HYBRID closure (Tiedtke, 1989)
ECMWF operational convection scheme (Bechtold et al., 2004)
with the shallow convection closure of Grant and Brown (1999)
ECMWF operational convection scheme (Bechtold et al., 2004) |

---

## Author Response (AR2)

**Chemical aging of atmospheric mineral dust during transatlantic transport**

**Reply to Anonymous Referee #2 (comments on the revised MS; doi:10.5194/acp-2016-470-RC1)**

by Mohamed Abdelkader, Swen Metzger, Benedikt Steil, Klaus Klingmüller, Holger Tost, Andrea Pozzer, Georgiy Stenchikov, Leonard Barrie, and Jos Lelieveld

January 29, 2017

We thank the referee for the additional comments, which are addressed below (in black).

*The authors have improved the quality of the manuscript, adhering to comments and suggestions that were made during the first review. Nevertheless, some clarifications and corrections are still needed in the text. My main comments are related to the model evaluation, the chemical aging sensitivity section and conclusions. I suggest Minor revision based on the following specific comments:*

*Page 4, line 20: the references to prior studies that reported interactions of organic acids with dust particles must not be omitted from the text. Any statement about "many modeling studies" must be supported by relevant citations.*

We apologize for this oversight. We have again included the references.

*Page 6, lines 30-35: It is unclear if the scatter plots in Fig.4 refer to all available stations or the six selected ones. I would expect that all stations are included, since this is the core of the long-term evaluation. If not, please include all available stations in the scatter plots.*

The scatter plot (Fig 4) indeed includes all available observation from all stations. We have added to the revised MS *"the corresponding scatter plots of both sides of the Atlantic Ocean include the observations from all stations"*.

*The same question applies to Table 2, even though from the title it seems that only the stations in Fig.3 are included in the calculation of the metrics. I am not sure I understand why all available stations are not included for the long-term evaluation. It appears as though all stations are used for the skill score (SS1) only. Please be consistent throughout the long-term evaluation section by using all available stations. If not, there has to be a clear reason and justification about which stations were selected to avoid the notion of cherry-picking in the selection of the stations.*

We apologize for inconsistency. Indeed, the statistics in Table 2 consistently includes all stations. We have added a note in the table caption.

*Fig.4, scatter plots: what are the dotted lines included in these plots (besides the obvious 1:1 line)?*

The lines shows the 1:2 and 1:10 ratios. We added a note in the figure caption.

*Page 6, line 33: The statement "simulated AOD compares well with the observations" does not come directly when one sees the scatter plots in Fig.4 where the model underestimates most of the AOD values and shows a large scatter in the West part. What is the measure for "well"? I suggest a more quantitative description of the results that is appropriate for a scientific paper. Statements that follow on page 7 and read "slightly underestimate" or "agree slightly better", "reasonably well", "generally somewhat underestimates" must be avoided. There are instances that the AOD is underestimated by more than a factor of 2 by the model. I suggest that the authors are specific and not descriptive.*

We have revised the MS accordingly.

*Page 7, line 2: the lower variability of the model is attributed to the model grid spacing without any support for this statement. Did the authors see a different variability when running a higher grid resolution? Prior work should also be referenced here if such information is not available from the current simulations, otherwise this statement is pure speculation.*

We have added a reference.

*Page 7, line 5 (and elsewhere): Throwing the values of the statistical metrics at the reader without any description of their meaning is not helping the discussion of the results. Each metric provides a different side of the model evaluation story and should be carefully narrated, and thus, explained. SS1 is 0.7 and 0.73 for the average of the selected stations but in Fig. 4 there are at least 8 stations with SS1 below 0.4. The interpretation of SS1 must be carefully described in the text.*

The statistical parameters are discussed on Pg 6 L4 L27. Skill score (SS1) is described in detailed in the supported reference (Pg 5 L10) in the revised MS. The stations are selected based on the availability of observations during the period 2000-2013. All stations with a low SS1 value have a much lower number of observations and are thus statistically less significant. We added an explanation in the revised MS.

*Page 15, lines 10-17: The "Aging" versus "no Aging" comparison is a very important outcome of this study. In order to provide context for the "improved statistics" the authors should provide the same statistical metrics as in Table 2 but for the "no aging" long-term simulation. The scatter plots in Fig.4 should also be provided for the "no aging" simulation. Such comparison will give a straightforward understanding on the advantages/benefits of using the suggested chemical aging approach.*

An deeper analysis of the "Aging" versus "no Aging" comparison for the long-term simulation is beyond the objective of this study and will be presented elsewhere.

*Related to the previous comment, Fig.12 does not reveal any significant improvement in the AOD with the Aging vs. No-Aging sensitivity test for 2009. The same applies to precipitation in Fig.15.*

A table of statistical metrics with all experiments is provided in the supplement (Pg 11). Our aging sensitivity study aims on scrutinizing the impact of the chemical aging of mineral dust in terms of AOD relative to other model parameters that control the transatlantic dust transport. However, the objective of this study was not to improve the EMAC model to an absolute maximum. To achieve this, first a deeper understanding of the general model behaviour is a pre-requisite. And by this work, we hope to have contributed to that goal, so that other studies can benefit from the insights obtained here.

[revised manuscript text omitted]